# A distinct mammalian disome collision interface harbors K63-linked polyubiquitination of uS10 to trigger hRQT-mediated subunit dissociation

Translational stalling events that result in ribosome collisions induce Ribosome-associated Quality Control (RQC) in order to degrade potentially toxic truncated nascent proteins. For RQC induction, the collided ribosomes are first marked by the Hel2/ZNF598 E3 ubiquitin ligase to recruit the RQT complex for subunit dissociation. In yeast, uS10 is polyubiquitinated by Hel2, whereas eS10 is preferentially monoubiquitinated by ZNF598 in human cells for an unknown reason. Here, we characterize the ubiquitination activity of ZNF598 and its importance for human RQT-mediated subunit dissociation using the endogenous XBP1u and poly(A) translation stallers. Cryo-EM analysis of a human collided disome reveals a distinct composite interface, with substantial differences to yeast collided disomes. Biochemical analysis of collided ribosomes shows that ZNF598 forms K63-linked polyubiquitin chains on uS10, which are decisive for mammalian RQC initiation. The human RQT (hRQT) complex composed only of ASCC3, ASCC2 and TRIP4 dissociates collided ribosomes dependent on the ATPase activity of ASCC3 and the ubiquitin-binding capacity of ASCC2. The hRQT-mediated subunit dissociation requires the K63-linked polyubiquitination of uS10, while monoubiquitination of eS10 or uS10 is not sufficient. Therefore, we conclude that ZNF598 functionally marks collided mammalian ribosomes by K63-linked polyubiquitination of uS10 for the trimeric hRQT complex-mediated subunit dissociation.

Ribosome stalling induces quality control pathways targeting the mRNA (NGD: No-Go Decay) and the nascent polypeptide (RQC: Ribosome-associated Quality Control). The RQC pathway monitors translation and ensures the efficient elimination of aberrant nascent protein products. Collided disomes or trisomes consisting of the leading ribosome and the following colliding ribosome(s) serve as a proxy for translation problems in the cell and are ubiquitinated by the E3 ubiquitin ligase Hel2 and the E2 ubiquitin-conjugating enzyme Ubc4 in yeast[1]. As a result, the RQC-Trigger (RQT) complex recognizes ubiquitinated collided ribosomes as a substrate to initiate the RQC pathway by ribosomal subunit dissociation. The yeast RQT complex is composed of the RNA helicase-family protein Slh1, the ubiquitin-binding protein Cue3, and Rqt4[1–3]. The ubiquitin-binding activity of Cue3 and the ATPase activity of Slh1 are responsible for the subunit dissociation of polyubiquitinated collided ribosomes both in vivo and in vitro[1,3]. Analogously, ZNF598, the human homolog of Hel2, ubiquitinates the ribosomal proteins uS10 and eS10 to initiate RQC in mammals[4–6]. The subsequent ribosomal subunit dissociation of collided ribosomes by the human homolog of the RQT complex termed hRQT or ASC-1

✉ e-mail: beckmann@genzentrum.lmu.de; toshiinada@ims.u-tokyo.ac.jp

complex points to a conserved role of ribosome ubiquitination in quality control[7,8].

It is believed that the distinct inter-ribosomal interface of collided ribosomes facilitates their recognition by Hel2/ZNF598, which thereby serves as a specific ribosomal collision sensor[9]. Cryo-EM structures of both the yeast and the rabbit disome as a minimal ribosomal collision unit showed that the general architecture of these disomes is conserved[9,10]. The leading ribosome is usually stalled in the classical POST-translocation state (dependent on the actual cause for stalling), while the second colliding ribosome is in a hybrid state with A/P and P/E-tRNAs, apparently locked in an incomplete translocation step. The interface between the leading and the colliding ribosome is mainly formed by the small subunits (40S), and to a lesser extent, also by the large ribosomal subunit (60S) of the leading ribosome.

Despite the progress made in recent studies, we lack direct structural information on human collided disomes as a minimal recognition unit to initiate RQC. In this regard, studies of endogenous stallers, such as poly(A) tracts and *SDD1* in yeast, which are subject to RQC, provided a valuable mechanistic understanding of ribosomal stalling and collisions[3,4,11–13]. Besides the poly(A) sequence, which is not genetically encoded, additional endogenous targets of the RQC surveillance system were identified. In yeast, the *SDD1* endogenous staller employs a combined ribosomal stalling mechanism and the trisome formed on the *SDD1*-stalling sequence is efficiently ubiquitinated by Hel2 and subjected to RQC[3]. In humans, the *XBP1u* mRNA precursor of the endoplasmic reticulum (ER)-stress-responsive transcription factor XBP1, was identified by disome profiling as a strong endogenous ribosomal collision-inducing mRNA[14]. In this study, the *XBP1u* mRNA was shown to cause queues of collided ribosomes, which are subjected to RQC[14].

Although both the yeast and the human RQC pathways and their respective initiation via Hel2/ZNF598 and RQT/hRQT seems to be conserved, there are many open questions. For instance, ZNF598 was proposed to preferentially monoubiquitinate the eS10, uS10 ribosomal protein in mammals[4–6], whereas yeast Hel2 functionally polyubiquitinates uS10[1,9]. The K63-specific polyubiquitin linkage-type, which was found to be crucial in yeast, was not reported in mammals. Following Hel2/ZNF598 ubiquitination, the Slh1/ASCC3 helicase subunit of the RQT complex preferentially dissociates the first stalled ubiquitinated ribosome in an ATP-dependent manner[3,8]. While both the ubiquitin-binding activity of Cue3 and the ATPase activity of Slh1 are required for the subunit dissociation in yeast[1,3], ASCC2, the homolog of Cue3, was proposed to have little to no functional role in humans[8]. In a contradicting finding, the ubiquitin-binding capacity of ASCC2 was suggested to be required for RQC[7]. Moreover, the composition of the human RQT (hRQT or ASC-1) complex is somewhat unclear since its components have another role in DNA repair in the nucleus[15]. In the context of RQC, the hRQT complex comprises three subunits homologous to the yeast RQT factors: the ASCC3/Slh1, the ASCC2/Cue3, and the TRIP4(ASC-1)/Rqt4. The human ASCC1 subunit of the ASC-1 complex is crucial in DNA repair but does not seem to be involved in the RQT activity[7,8].

Here we show, that ZNF598 functionally marks collided mammalian ribosomes by K63-linked polyubiquitination of uS10 for the trimeric hRQT complex-mediated subunit dissociation. We reconstitute the ZNF598-mediated collision-dependent polyubiquitination of stalled ribosomes and the subsequent hRQT complex-mediated subunit dissociation, using the endogenous *XBP1u* and poly(A) stalling sequences. We structurally characterize a human disome collided on the *XBP1u* mRNA and reveal two additional inter-ribosomal contact sites not described before for mammalian disomes. Moreover, we observe a striking difference in the arrangement of human and yeast collided disomes, which could explain their different ubiquitination patterns. Our data reveal that ZNF598 forms K63-linked polyubiquitin chains on uS10, and, importantly,

the hRQT-driven splitting of ribosomes collided on endogenous *XBP1u* and poly(A) staller mRNAs is only detectable for ribosomes marked with the K63-linked polyubiquitinated uS10. Mono-ubiquitination of eS10 or uS10 is not sufficient for detectable sub-unit dissociation by the hRQT complex. Moreover, we observe that ASCC2 specifically interacts with K63-linked polyubiquitin chains and show that mutations in the ubiquitin-binding domain of ASCC2 disrupt the hRQT activity. These results demonstrate the conserved mechanism of RQC initiation in eukaryotes.

## Results

### Polyubiquitination of uS10 by ZNF598 is conserved and takes place upon ribosomal collision both in vitro and in vivo

Ubiquitination of collided ribosomes is required for the RQT complex-mediated subunit dissociation as a prerequisite of RQC. Therefore, we investigated the ubiquitination mode of ribosomal proteins by ZNF598 upon ribosomal stalling and collision by reconstituting the translational arrest with various stalling mRNAs in vitro using the rabbit reticulocyte lysate system (RRL; Fig. 1A, B). As expected, all mRNA constructs lead to ribosome stalling and ribosome-nascent chain complex (RNC) formation, as indicated by the presence of peptidyl-tRNA arrest products (Fig. 1C). The sucrose density gradient ultracentrifugation of purified RNCs showed that for the endogenous staller *XBP1u*, the poly(A) stretch, and the stable Stem-loop structure mRNAs, the arrest product primarily co-migrated with polysomes, indicating stable ribosome collisions. In contrast, the *XBP1u-W256A* mutant mRNA, which disrupts stalling[16], the truncated mRNA, and the self-cleaving ribo-zyme (Rz) mRNA almost exclusively generated stalled monosomes (Fig. 1D, E). This may be explained by the efficient recycling of these 80S ribosomes stalled at the end of the mRNA with an empty A-site by the DOM34/HBS1/ABCE1 system[17–19]. To assess the capability of ZNF598 to ubiquitinate any of these stalled ribosomes, we supplemented the in vitro translation reactions with recombinantly purified ZNF598 (Supplementary Fig. 1A) and isolated the resulting RNCs (Fig. 1B). Consistent with the previous reports[4], we find that both uS10 and eS10 were monoubiquitinated in a ZNF598-dependent manner (Fig. 1F). However, while ZNF598 primarily mono- and di-ubiquitinates ribosomal protein eS10[10], we also found that uS10 is polyubiquitinated by ZNF598 analogously to yeast. In particular, substantial ubiquitination was only observed for the *XBP1u*, the poly(A), and the Stem-loop constructs, affirming the requirement of ribosome collisions for ZNF598 activity. Correspondingly, ribosomal protein ubiquitination was not observed for uS10 and only to a lesser extent for eS10 in RNCs stalled on the truncated mRNA and the ribozyme mRNA.

To validate the relevance of our in vitro findings, we decided to monitor uS10 and eS10 ubiquitination in collided ribosomes in vivo. Therefore, we induced ribosome collisions by treatment with moderate concentrations of anisomycin in HEK293T human cells expressing either 3HA-tagged uS10 or eS10[4,10,11,20]. As a result of ribosomal collisions induced by mild anisomycin treatment, the ubiquitination levels of both uS10 and eS10 increased in treated lysates, showing not only mono- or di- but also polyubiquitination (Supplementary Fig. 1B, C). Accordingly, expression of uS10 or eS10 mutants lacking their respective ubiquitinated lysine residues (uS10-K4RK8R-3HA and eS10-K138RK139R-3HA) (Supplementary Fig. 1B, C), abolished their ubiquitination. Interestingly, we observed background of monoubiquitinated eS10 even without anisomycin treatment (Supplementary Fig. 1C). Taken together, these results show that the ZNF598-dependent ubiquitination of ribosomal proteins uS10 and eS10 is conserved from yeast to human and takes place both in vitro and in vivo upon ribosomal collision. Moreover, these results suggest that polyubiquitination of ribosomal protein uS10 in mammals is collision-dependent and might play a similar role as in yeast.

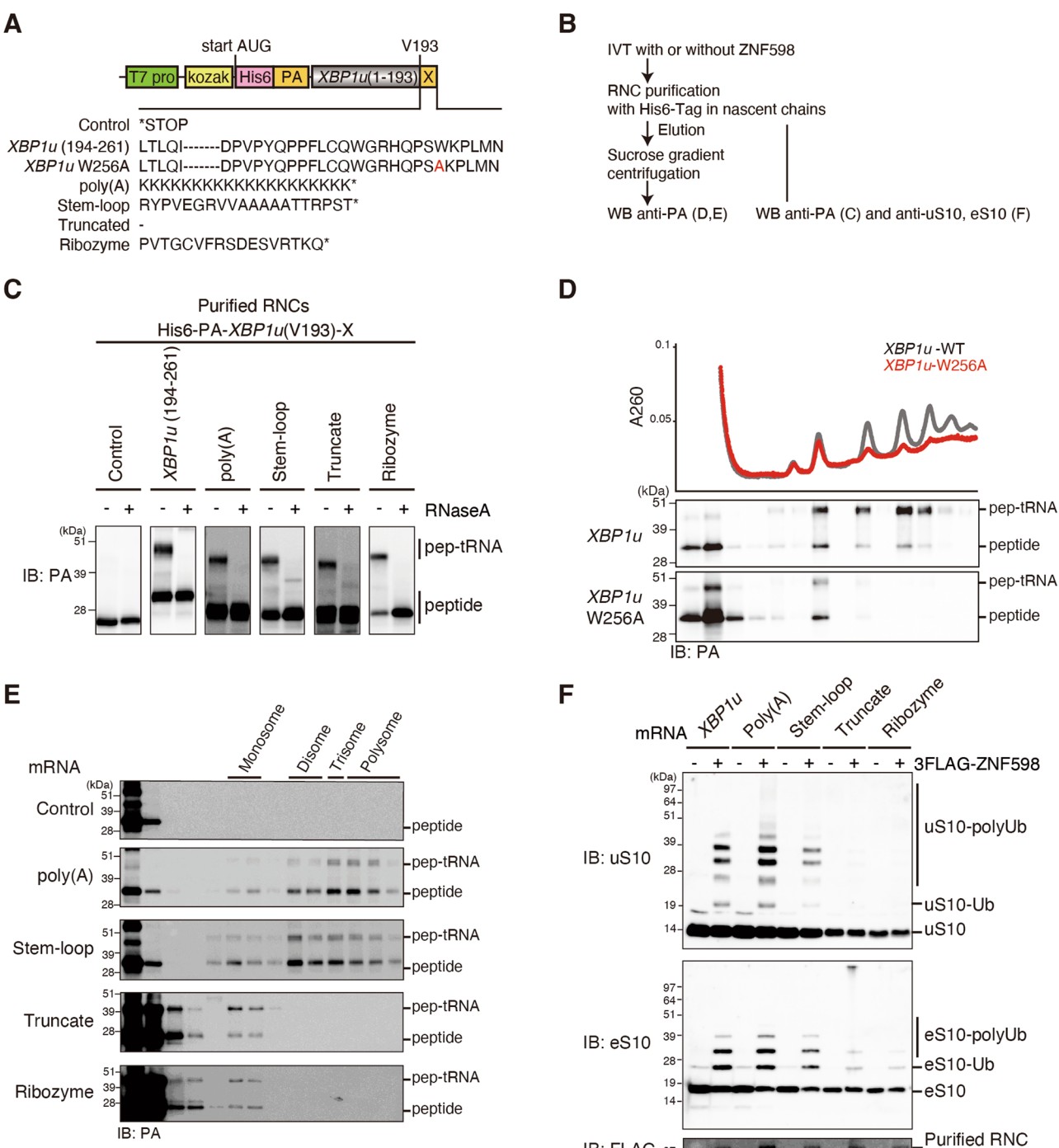

**Fig. 1 | In vitro reconstitution of translational arrest and ZNF598-mediated ubiquitination. A** Schematic representation of mRNA templates used in RRL in vitro translation reactions. All constructs encode the N-terminal part of XBP1u (1–193), which is appended (X) by either the *XBP1u* (194–261) stalling sequence, a poly(A) stretch, a stable RNA stem-loop sequence or a self-cleaving ribozyme (Rz). All constructs contain an N-terminal His$_6$ and PA-tag for affinity purification and antibody detection. **B** Schematic overview of the in vitro stalling and ubiquitination assays. **C–E** In vitro translation of staller mRNAs from **A**. The free peptide and the peptidyl-tRNA (pep-tRNA) arrest products were detected by Western blotting using an anti-PA antibody. We obtained essentially the same results as two independent

experiments. **C** Purified RNCs were treated with RNase A to verify the peptidyl-tRNA arrest products. **D**, **E** RNCs purified from in vitro translation reactions using all mRNA templates (**A**) but *XBP1u (1–235)* is used as a control and were subjected to ultracentrifugation through sucrose density gradients and individual gradient fractions were visualized by Western blotting using the anti-PA antibody. **F** In vitro ubiquitination by ZNF598 of RNCs stalled on mRNAs from **A**. RNCs from in vitro translation reactions with or without recombinantly purified 3FLAG-ZNF598 were purified and eS10 and uS10 ubiquitination was visualized by Western blotting. We obtained essentially the same results in at least three independent experiments.

## Collided human ribosomes create a specific composite interface

It is believed that the composite inter-ribosomal collision interface serves as a universal recognition cue for Hel2/ZNF598 irrespective of the particular cause of translation stalling[9,10]. Moreover, the use of

collided ribosomes as a proxy to cell response mechanisms seems to be conserved from bacteria to mammals, corresponding to the highly conserved nature of the ribosome itself[9,10,21]. However, direct structural information on human collided disomes as a minimal recognition unit

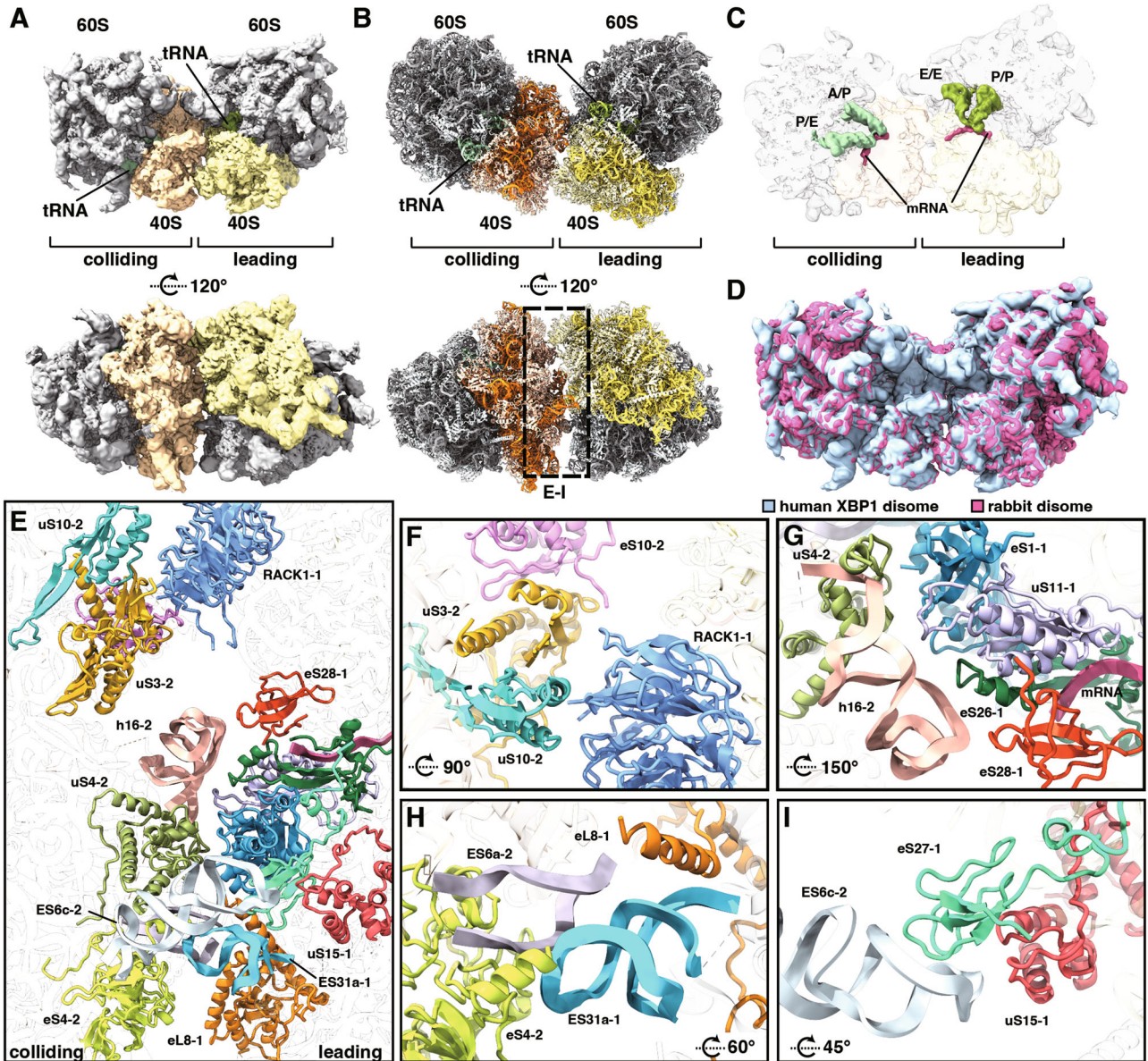

**Fig. 2 | Cryo-EM structure comparison of the human XBP1-stalled disome and the yeast disome. A** Composite cryo-EM density and the molecular model (**B**) of the human XBP1-stalled collided disome. **C** Cut-in view of the tRNA states. The leading ribosome is in a non-rotated POST-state with P/P E/E-tRNAs and an empty A-site. The colliding ribosome is in a rotated PRE-translocation state with A/P P/E hybrid tRNAs. **D** Cryo-EM structures of the human XBP1-stalled disome and the rabbit eRF1 mutant stalled disome (reconstructed from individual maps EMD-0192, 0194, 0195, and 0197[10]) were superimposed. Little variance in the inter-ribosomal orientation is observed between the two disomes. **E** Overview of the inter-ribosomal interface between leading and colliding ribosome with orientation indicated in panel **B**. **F–I** Focused views on main contact areas with indicated rotation in reference to panel **E**. **F** Detail of 40S head-to-head contacts. **G** Detail of the 40S body-to-body contacts around 18S rRNA h16 of the colliding ribosome (h16-2). **H** Detail of the interactions between the 60S of the leading ribosome and the 40S of the colliding ribosome. **I** Detail of the contacts of the 18S rRNA expansion segment ES6c of the colliding ribosome (ES6c-2) with the 40S proteins of the leading ribosome. Focused view panels with indicated rotations refer to orientation in lower panel **B**.

to initiate RQC were missing, since mainly the rabbit in vitro translation system has been used to study mammalian co-translational quality control pathways. Therefore, we decided to structurally characterize the collided human disome as a substrate for ZNF598-mediated ubiquitination. We employed a cell-free translation extract derived from HeLa cells and translated the *XBP1u* staller mRNA optimized for stable stalling (S255A)[16] and subsequent purification of stalled RNCs via the tagged nascent chain (Supplementary Fig. 2). After purification, we segregated ribosomal species via ultracentrifugation through a sucrose density gradient and harvested the disome peak for cryo-EM analysis. Cryo-EM data were processed using the 80S extension approach as described previously (Supplementary Fig. 3)[3,9].

Accordingly, A/P P/E 80S classes with the most prominent neighboring ribosome density (3.9% of all particles) were used to generate a consensus refinement of the disome which was then further refined with a focus on the individual 80S ribosomes. The individually refined leading and colliding ribosomes were fitted into the consensus refinement to generate a composite map. We reached overall resolutions of 2.9 Å (leading) and 3.2 Å (colliding) for individual ribosomes and 3.0 Å for the composite map (Fig. 2A and Supplementary Fig. 4). This allowed us to fit and refine molecular models to describe the collision interface (Fig. 2B and Supplementary Table 1).

All the available yeast and rabbit cryo-EM disome structures show the leading ribosome stalled in the classical POST-translocation state

with an empty A-site and occupied P- and E-sites (P/P E/E). Except for the poly(A) stalled yeast disome[12], the second colliding ribosome is in a hybrid state with A/P and P/E-tRNAs, apparently blocked from completing the translocation step[9,10]. In the structure of the collided human disome, we also observe this standard arrangement (Fig. 2C), including details such as missing mRNA density at the inter-ribosomal interface due to the aforementioned flexibility. This overall architecture corresponds to the previously observed state of 80S ribosomes stalled by XBP1[22] and the universal nature of ribosome collision in mammals. This conserved architecture is further illustrated by only very small positional differences we observed when comparing the structures of rabbit and human disomes (Fig. 2D). Moreover, we also observed two inter-ribosomal contact sites analogous to the ones described in the rabbit disome (Fig. 2E). The head-to-head contact site comprises RACK1 of the leading ribosome (RACK1-1) and uS3-2, uS10-2, and eS10-2 of the colliding ribosome (Fig. 2F). All these proteins have been implicated in ribosomal stalling, collision, and subsequent quality control pathways[23–25] with uS3, uS10, and eS10 representing the primary ubiquitination targets of Hel2/ZNF598[1,4,5]. The mRNA contact site comprises proteins at the mouth of the mRNA exit channel of the leading ribosome closely juxtaposed to the mRNA entry channel of the colliding ribosome (Fig. 2G). In particular, this involves the uS4 protein and the 18S rRNA helix16 of the colliding ribosome (eS4-2, h16-2) and the eS1, uS11, eS26, and eS28 proteins of the leading ribosome (eS1-1, uS11-1, eS26-1, and eS28-1; Fig. 2G).

Surprisingly, we observed two additional contact sites, which were not described for the rabbit disome structure[10]. First, we observed a 60S-40S contact comprising 18S rRNA expansion segment ES6a and the eS4 protein of the colliding ribosome (ES6a-2, eS4-2) and the 28S rRNA expansion segment ES31a and the eL8 protein of the leading ribosome (ES31a-1, eL8-1; Fig. 2H). Interestingly, there is no density for this contact in the cryo-EM multibody map of the rabbit disome (EMD-0195, 0197). Moreover, we observed a contact of the 18S rRNA expansion segment ES6c of the colliding ribosome (ES6c-2) reaching over to the eS15 and eS27 proteins of the leading ribosome (eS15-1, eS27-1; Fig. 2I). This is in agreement with a corresponding density also present in the multibody refined map of the rabbit disome. Furthermore, all the rRNA segments involved in contact are well conserved between rabbit and human. Taken together, we propose that the human and rabbit collided disomes are largely equivalent and that the easily accessible rabbit translation system may thus indeed serve as a model for studying collision-dependent human quality control pathways. Nonetheless, the additional contact sites we describe here significantly contribute to the overall architecture of mammalian collided disomes.

## Non-uniform architecture of eukaryotic ribosome collisions

While we found minor differences between the overall assemblies of the human and rabbit collided disomes, we observed striking divergence when comparing the structure of human and yeast collided disomes. In the yeast disomes the position of the colliding ribosome is rotated by 18° along an axis longitudinal to its 40S subunit from head to foot respective to the human colliding ribosome (Fig. 3A). This is caused by the distinct interaction between the 40S of the colliding ribosome and the 60S of the leading ribosome (40S–60S contact) in yeast. While the eS4-2 contact to the 28S rRNA ES31a-1 is analogous in both disomes, the 18S rRNA ES6a-2 contact to eL8-1 is missing in yeast (Fig. 3B, C). This is probably caused by a shorter C-terminal helix of the yeast eL8 protein in comparison with the human eL8 (Fig. 3C). More importantly, there is an alternative 40S–60S contact not present in human disomes[9]. This alternative contact is between the expansion segment ES6b of the colliding ribosome (ES6b-2) and the eL27-1 protein of the leading ribosome (eL27-1; Fig. 3D, E). It represents one of the main anchor points in the yeast disome, setting the distance of the colliding ribosome to the leading ribosome in this area. As a result,

while the mRNA exit face of the collided ribosome appears rotated towards the leading ribosome and makes closer, more stable contact, the opposite side appears shifted further away (Fig. 3A, F). This is projected to the head-to-head contact site, where both Asc-1 (RACK1) proteins are in close contact in the yeast disome (unlike in the human disome), the position of uS10-2 seems to remain relatively unchanged, while the position of eS10-2 seems to rotate further away from the leading ribosome when compared to the human disome (Fig. 3G). As mentioned above, this head-to-head contact area is believed to represent the main signaling hub for subsequent quality control pathways, with RACK1 being crucial for ribosomal collision[23–25] and uS10 representing the primary ubiquitination target of Hel2 in yeast[1]. The ubiquitination of eS10 for stalling by poly(A) is rather mammalian specific[4–6], and the proposed crucial role of eS10 ubiquitination in RQC and the ubiquitination of eS10 have no evidence in yeast[1,3,9]. This striking difference in the disome architecture may explain the observed differences in ubiquitination, higher stability and smaller hinging movements in the yeast disome. More importantly, this increased stability could contribute to the recognition and effectivity of downstream quality control pathways.

## The hRQT complex dissociates uS10 polyubiquitinated ribosomes collided on endogenous stallers

Despite the structural differences between mammalian and yeast disomes, we confirmed a common principle in the structural organization of ubiquitination targets of Hel2/ZNF598 at the inter-ribosomal interface. Moreover, we observed the collision-dependent polyubiquitination of uS10 in mammals which suggests that it might play a similar role as in yeast. Therefore, we further investigated the ubiquitination status of ribosomal proteins in the RQT-mediated subunit dissociation in mammals. To generate collided ribosomes as a ZNF598 and RQT substrate, we used the endogenous XBP1u and poly(A) stalling mRNAs in RRL in vitro translation reactions (Fig. 4A, B). Both these stallers were shown to induce RQC in mammals[4,11,14], and we confirmed that both induce ribosome collisions (Figs. 1, 4C, E). The RNCs were isolated via the His_6-tagged nascent chain and ubiquitinated by ZNF598 while tethered to magnetic beads. After elution, the RNCs were incubated with either ATP alone or with added purified hRQT complex (ASCC3, ASCC2, and TRIP4) in order to monitor subunit splitting (Supplementary Fig. 5A). After the reaction, samples were subjected to sucrose density gradient ultracentrifugation, and fractions of sucrose gradients were analyzed by Western blotting with anti-uS10 (Fig. 4C–F, top panels) and anti-eS10 antibodies (Fig. 4C–F, middle panels). In the control reactions without the hRQT complex, we found that ZNF598 forms polyubiquitin chains on both uS10 and eS10 (Fig. 4C for XBP1u; Fig. 4E for poly(A)). This again applied to both the XBP1u and the poly(A) mRNAs, similarly to what we found before (Fig. 1F). However, the polyubiquitination was predominantly observed in the di-, tri- and polysome fractions and not detectable in the 40S fraction (Fig. 4C for XBP1u; Fig. 4E for poly(A)). In reactions with added purified hRQT complex, we observed subunit dissociation and polyubiquitinated uS10 was detected in the 40S fraction while very faint ubiquitin bands of ubiquitinated eS10 was detected (Fig. 4D for XBP1u; Fig. 4F for poly(A)). Finally, we wanted to distinguish the subunit products of the hRQT complex-mediated dissociation from subunits present in the RNC input. Therefore, we used the heavy polysome fraction (trisomes and heavier) separated from RNCs primed with the XBP1u stalling mRNA, purified from the RRL in vitro translation reaction and ubiquitinated by ZNF598 in the hRQT-mediated splitting reaction (Supplementary Fig. 6A–D). Again, polyubiquitinated uS10 was only observed in the resulting 40S fractions after incubation with the hRQT complex (Supplementary Fig. 6C, D). This indicates that the hRQT complex dissociates the polysomes marked with polyubiquitinated uS10 into subunits.

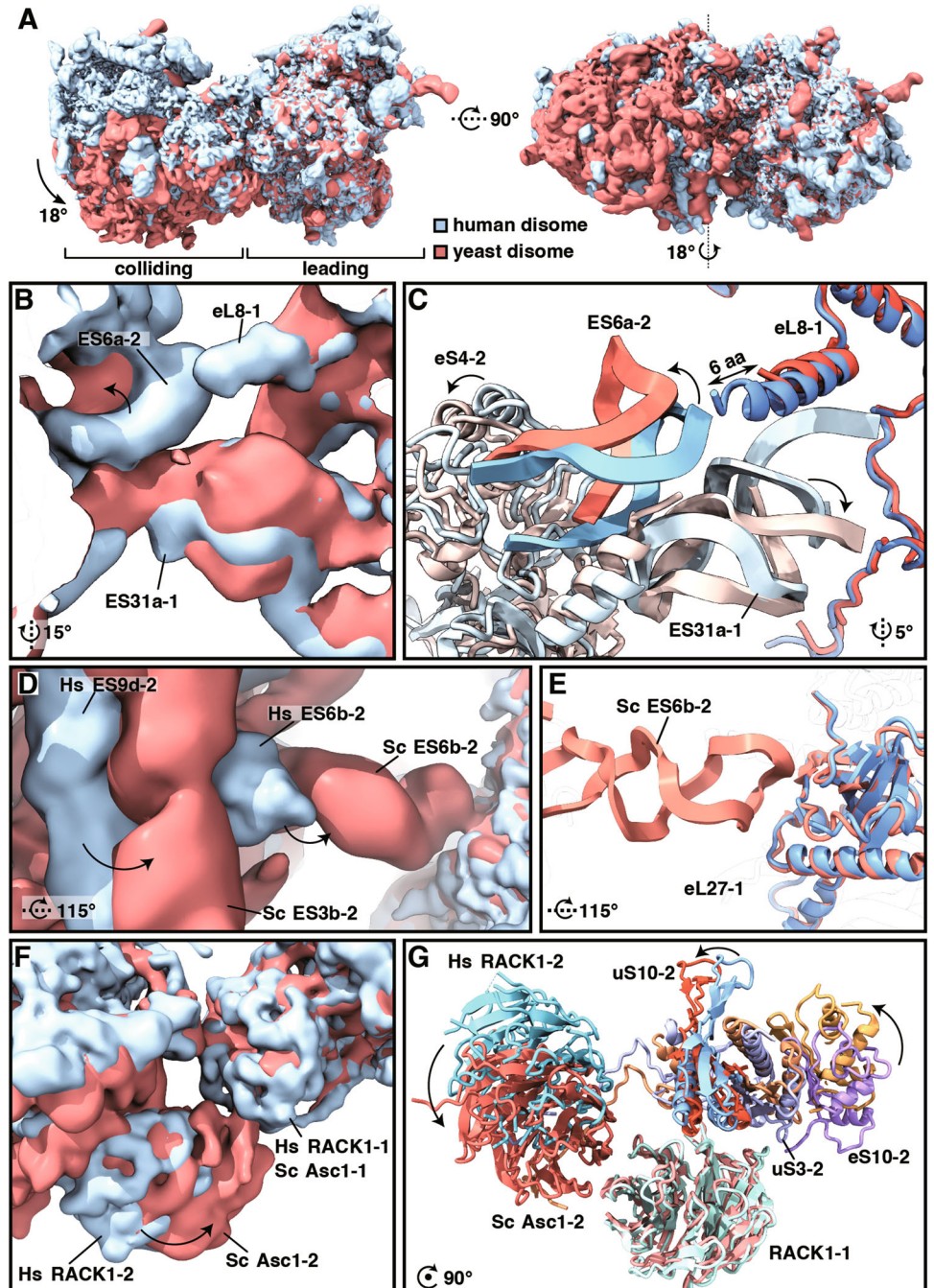

**Fig. 3 | Structural comparison of the human and the yeast disome.** Cryo-EM densities and models are colored in shades of blue for the human disome and red for the yeast disome (EMD-4427 [https://www.ebi.ac.uk/emdb/EMD-14181], PDB ID: 6I7O [https://doi.org/10.2210/pdb7QVP/pdb]) in all panels. Focused views with indicated rotations refer to panel **A** left. **A** Overlay of the cryo-EM densities of the human and the yeast disomes shows an apparent 18° rotation of the colliding ribosome with respect to the leading ribosome in yeast compared to the human disome. **B**, **C** Detailed view of the single human inter-ribosomal 40S–60S contact area. **D**, **E** Detail of the yeast-specific inter-ribosomal 40S–60S contact via the expansion segment ES6b of the colliding ribosome (ES6b-2). **F**, **G** Detail of differences in the head-to-head contact area.

Importantly, this experimental setup allowed us to observe the subunit dissociation products together with a clear decrease in the amount of polysomes (Supplementary Fig. 6C, D top panels).

Together, these results suggest that the polyubiquitination of uS10 on collided ribosomes is crucial for splitting by the hRQT complex. Furthermore, the ZNF598-dependent ubiquitination and subsequent subunit dissociation by the hRQT complex targets ribosomes collided on different endogenous stallers in the same way, as observed for the *XBP1u* and the poly(A) staller constructs.

## The hRQT activity requires K63-linked polyubiquitination, ASCC3 ATPase, and ASCC2 ubiquitin-binding activities

We demonstrated that particularly uS10 polyubiquitinated small ribosomal subunits are produced by the hRQT complex-mediated subunit dissociation. In yeast, the K63-linked polyubiquitination of uS10 is specifically required to induce RQC[1,9]. Therefore, we investigated which polyubiquitin chain type is formed by ZNF598 and whether it is important for the hRQT activity in mammals. To determine which kind of ubiquitin linkage is formed by ZNF598 on uS10 and eS10,

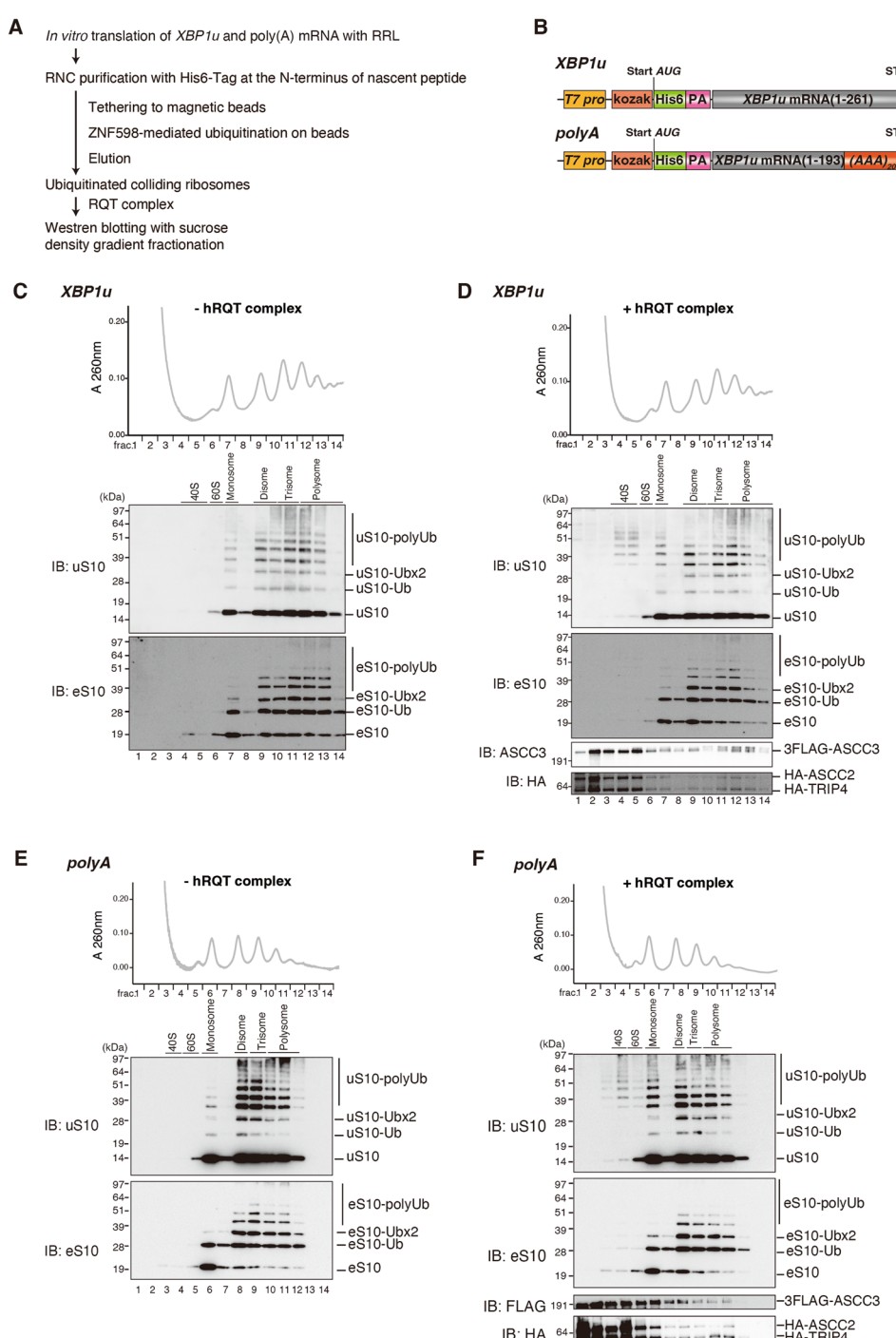

**Fig. 4 | hRQT-mediated subunit dissociation of collided ribosomes. A** Schematic overview of the hRQT ribosome splitting assay. **B** Schematic representation of the *XBP1u* and poly(A) staller mRNAs. **C**–**F** Collided ribosomes derived from in vitro translation reactions of *XBP1u* (**C**, **D**) or poly(A) (**E**, **F**) staller mRNAs were isolated and ubiquitinated by ZNF598. The purified ubiquitinated RNCs were incubated with (**D**, **F**) or without (**C**, **E**) the hRQT complex. Ribosomal species resulting from the splitting reaction were separated by ultracentrifugation through sucrose density gradients. Gradient fractions were analysed by Western blotting using antibodies against ribosomal proteins uS10 and eS10 and against HA- and FLAG-tag. We obtained essentially the same results in at least three independent experiments.

we generated and purified collided RNCs using the *XBP1u* staller mRNA in vitro (Fig. 5A). We then collected the polysome fractions after ultracentrifugation through a sucrose density gradient (Fig. 5B). The obtained collided ribosomes were in vitro ubiquitinated by ZNF598 using either wild-type or mutant ubiquitin variants.

We confirmed that uS10 is polyubiquitinated with wild-type ubiquitin (Ub-WT) (Fig. 5C, upper panel, comp. Figs. 1F, 4C–F), while the control without ubiquitin showed no ubiquitination. When we used a

ubiquitin mutant that contains only K63 and no other lysine residues (Ub-K63only), the amount of uS10 polyubiquitination was decreased but still significant (Fig. 5C, upper panel). Moreover, the overall ubiquitination pattern appeared identical to the one observed with Ub-WT, strongly suggesting that ZNF598 naturally forms K63-linked ubiquitin chains on collided ribosomes. In line with this, we observed only two ubiquitination bands for uS10 when using Ub-K63R, a ubiquitin version that contains a K63R mutation or Ub-K0 that contains no

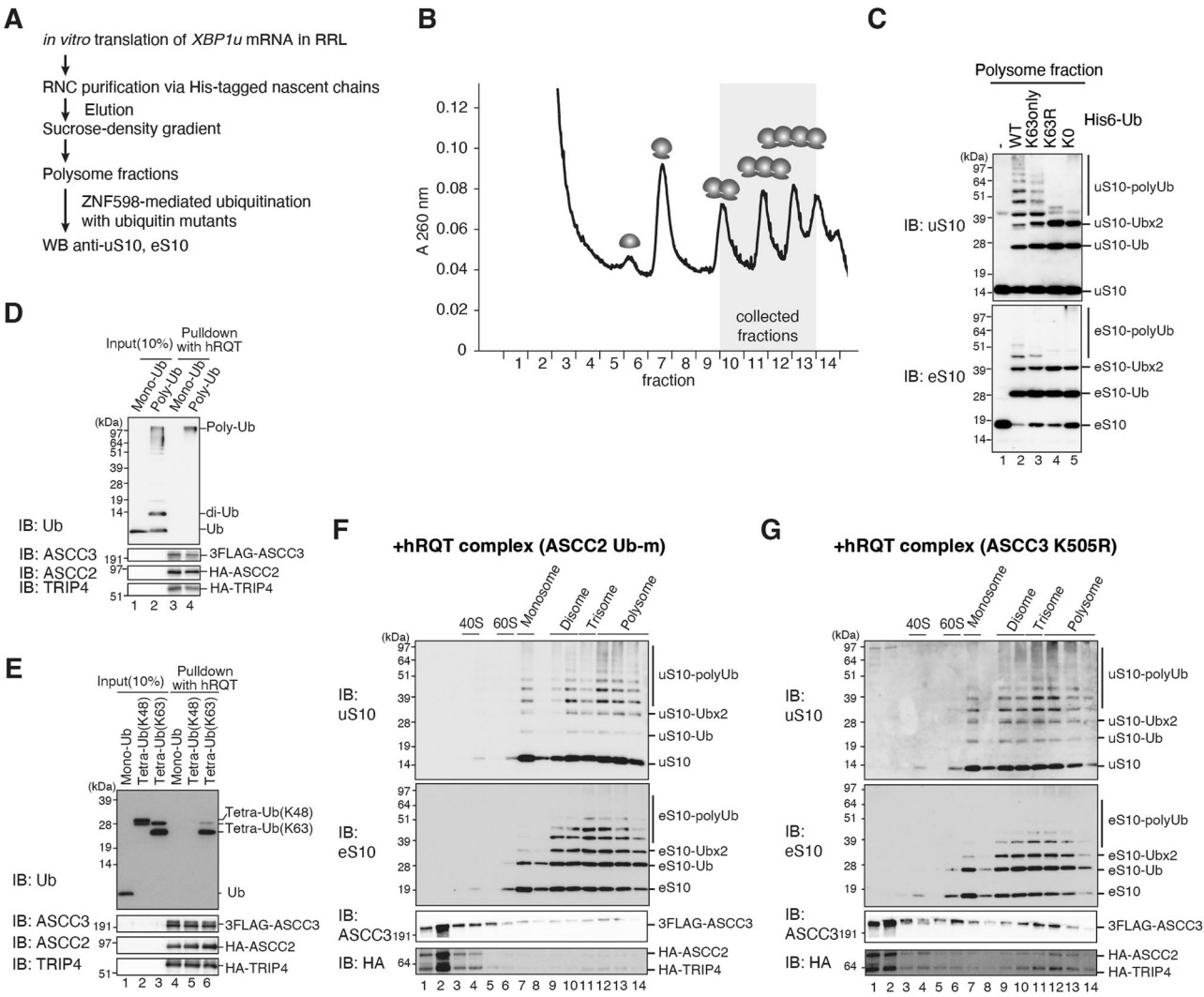

**Fig. 5 | The hRQT complex activity requires ASCC2 binding to K63-linked polyubiquitin chains and ASCC3 ATPase activity. A** Schematic overview of the ZNF598-mediated in vitro ubiquitination assay using *XBP1u*-stalled collided ribosomes. **B** Purified *XBP1u*-RNCs were separated by sucrose density gradient ultracentrifugation. Indicated polysome fractions (di-, tri-, and tetrasomes) were collected for the ZNF598-mediated ubiquitination reaction. **C** The collected polysome fractions were ubiquitinated by ZNF598 using either His₆-tagged wild-type ubiquitin (WT-Ub) or the indicated mutants (K63only-Ub, K63R-Ub, and K0-Ub). Reactions were analysed by western blotting with anti-uS10 and -eS10 antibodies. We obtained essentially the same results in at least three independent experiments. **D, E** In vitro binding assay of K63-linked polyubiquitin chain with the hRQT complex. The hRQT complex composed of FLAG-ASCC3, HA-ASCC2, and HA-TRIP4 on the beads were incubated with monoubiquitin or K63-linked polyubiquitin chain

(**D**), and monoubiquitin (Mono-Ub), K48-linked tetraubiquitin chains (Tetra-UB(K48)), or K63-linked tetraubiquitin chains (Tetra-Ub(K63)) (**E**). Samples were analysed by western blotting with anti-Ub, -ASCC3, -ASCC2, or -TRIP4 antibodies. We obtained essentially the same results in three independent experiments. **F, G** In vitro ribosome splitting assay with the mutant hRQT complex containing either the ubiquitin-binding impaired ASCC2-Ubm (**F**) or the ATPase-deficient ASCC3-K505R (**G**). RNCs generated from *XBP1u* staller mRNA were purified, ubiquitinated by ZNF598, and incubated with the particular mutant hRQT complex. These hRQT in vitro splitting reactions were then subjected to ultracentrifugation through sucrose density gradients and gradient fractions were analysed by western blotting using indicated antibodies. We obtained essentially the same results in at least three independent experiments.

lysine residues at all. The lower band corresponds to monoubiquitinated uS10 (uS10-Ub), while the upper band likely represents monoubiquitination at the two target residues of uS10 (uS10-Ubx2). This is the only plausible explanation in the case of the Ub-K0 mutant, which does not allow for any linkage between ubiquitin molecules. Analogously, we found that eS10 is also only polyubiquitinated by ZNF598 with the Ub-WT and the Ub-K63only variants (Fig. 5C, lower panel). However, eS10 was mainly monoubiquitinated and only minor polyubiquitination was detected in comparison with uS10.

Since the hRQT complex component ASCC2 contains a ubiquitin-binding CUE domain, we examined the K63-linked polyubiquitin binding capability of the hRQT complex. To test this, we incubated ubiquitin with ATP and the heterodimeric E2 ubiquitin-conjugating

Ubc13-Mms2 complex to in vitro synthesize K63-linked polyubiquitin chains[26] (Supplementary Fig. 5C). We confirmed the generation of K63-linked polyubiquitin chains by Western blotting using anti-K63-linked polyubiquitin chain antibody (Supplementary Fig. 5D). After binding of purified hRQT complex to antibody-coupled beads, we added the artificial ubiquitin chains or monoubiquitin to evaluate the ubiquitin binding. We found that the hRQT complex indeed binds the K63-linked polyubiquitin chain rather than monoubiquitin (Fig. 5D and Supplementary Fig. 5D). To assess whether the hRQT ubiquitin-binding activity is indeed linkage-specific, we performed a binding assay with antibody-coupled hRQT complex using mono- and K48- or K63-linked tetraubiquitin. In this setup, the hRQT complex was associated only with the K63-linked tetraubiquitin (Tetra-Ub(K63)) but not with the

K48-linked tetraubiquitin (Tetra-Ub(K48)) or with monoubiquitin (Mono-Ub) (Fig. 5E). This indicates that the hRQT complex specifically binds to K63-linked polyubiquitin chains.

We further asked whether the ubiquitin-binding activity of ASCC2 is required for the hRQT-mediated ribosome subunit dissociation, as this was disputed in the literature[7,8]. To test this, we generated collided, ubiquitinated ribosomes using the *XBP1u* staller mRNA in vitro (Fig. 4A). The obtained RNCs were subjected to ultracentrifugation through a sucrose density gradient after incubation with a mutant hRQT complex containing a ubiquitin-binding deficient ASCC2 (ASCC2-Ubm) that was purified from ASCC2-knock down cells[7] (Supplementary Fig. 5B). This time, we did not observe any ribosome splitting as no polyubiquitinated uS10 was present in the 40S fraction (Fig. 5F). Moreover, the portion of the mutant hRQT complex co-migrating with polysomes was reduced compared to the wild-type hRQT complex (compare with Fig. 4D, F). Taken together, these results suggest that the capability of ASCC2 to bind ubiquitin is essential for ribosome recruitment and subsequent splitting by the hRQT complex.

In yeast, the ribosome subunit dissociation of collided *SDD1* trisomes by RQT also requires the ATPase activity of the Ski2-like helicase 1 (Slh1)[3]. Similarly, the ASC-1 complex requires the ATPase activity of ASCC3 in mammalian cells[7,8]. Therefore, we investigated the requirement of the ASCC3 ATPase activity for ribosome splitting in the context of the hRQT complex. To test this, we used a mutant hRQT complex containing the ASCC3-K505R ATPase activity deficient variant (Supplementary Fig. 5B). This mutant complex was used to disassemble polyubiquitinated collided ribosomes generated on the *XBP1u* mRNA in vitro (Fig. 4A). In contrast to the wild-type hRQT complex, no polyubiquitinated uS10 40S product of ribosomal splitting was detected (Fig. 5G). This indicates that the ASCC3-K505R containing complex is inactive. Considering these results, we conclude that the hRQT activity relies both on ASCC2, which specifically recognizes the K63-linked polyubiquitin chain formed by ZNF598 and on the ATPase activity of ASCC3.

### hRQT specifically dissociates ribosomes marked with K63-linked polyubiquitinated uS10

We then investigated whether K63-linked polyubiquitin chains formed by ZNF598 on collided ribosomes indeed define the substrate for the hRQT-mediated subunit dissociation. To test this, we ubiquitinated *XBP1u*-stalled collided RNCs with purified ZNF598 in vitro (Fig. 4A), using the Ub-K63R or the Ub-K63only mutants. Subsequently, these ubiquitinated RNCs were either first incubated with the hRQT complex or directly subjected to ultracentrifugation through a sucrose density gradient. No uS10 or eS10 polyubiquitination was detectable in the Ub-K63R mutant condition (Fig. 6A). Instead, both uS10 and eS10 were again monoubiquitinated on a single lysine (uS10-/eS10-Ub) or on both target lysine residues (uS10-/eS10-Ub×2) by ZNF598 (Fig. 6A and comp. Fig. 5C). No ubiquitinated uS10 or eS10 was detected in the 40S fractions after incubation of these monoubiquitinated RNCs with the purified hRQT complex (Fig. 6B, upper and middle panel). This indicates that the monoubiquitination of either eS10 or uS10 is not sufficient to induce ribosome splitting by the hRQT complex. Moreover, the hRQT complex did not co-migrate with monoubiquitinated ribosomes in the sucrose density gradient (Fig. 6B, lower panels), which is consistent with the K63-linked polyubiquitin binding specificity of ASCC2. As expected, polyubiquitination of uS10 and eS10 by ZNF598 was restored when the Ub-K63only ubiquitin mutant was used instead (Fig. 6C, upper and lower panels). Again, di-, tri-, and polysomes were preferentially polyubiquitinated by ZNF598, while only a minor polyubiquitination of monosomes was observed (comparable to Ub-WT presented in Fig. 4C, E). When we incubated these K63-linked polyubiquitinated collided ribosomes with the purified hRQT complex, we detected the polyubiquitinated uS10 in the 40S subunit dissociation fraction (Fig. 6D, upper panel). However, we did not detect any

polyubiquitinated eS10 in this 40S subunit dissociation fraction (Fig. 6D, middle panel). Additionally, the hRQT complex co-migrated with collided ribosomes throughout the gradient again (Fig. 6D, lower panels), underlining the importance of K63-linked polyubiquitination for its activity.

Taken together, these results confirm that K63-linked polyubiquitin chains formed by ZNF598 on collided ribosomes indeed mark the substrate for hRQT activity. In contrast to previous reports[4–6], monoubiquitination of eS10 or uS10 appears to be insufficient for both the hRQT substrate recognition and the subunit dissociation activity. Since only the polyubiquitinated uS10 but not eS10 is detectable in the subunit product fraction of hRQT activity, it is also possible that uS10 polyubiquitination specifically marks the substrate of hRQT.

## Discussion

The RQC pathway is a highly conserved quality control system in eukaryotes regarding both the factors required and the ubiquitination of the collided ribosomes as a general RQC initiation step. In yeast, Hel2-mediated K63-linked polyubiquitination of collided ribosomes drives the RQT-mediated subunit dissociation. In contrast, in mammalian RQC initiation, ZNF598 was suggested to monoubiquitinate ribosomal proteins in ribosomal collisions caused by inhibited translation termination or by a moderate treatment with elongation inhibiting antibiotics[5,6,10]. Furthermore, the mono- and di-ubiquitination of mammalian ribosomal proteins was reported for ribosomes stalled on the endogenous poly(A) staller[4]. To further dissect this, we started with the comparison of multiple stalling sequences and monitored the ubiquitination status of the resulting collided ribosomes. Surprisingly, we observed polyubiquitination of ribosomal protein uS10, which was ribosome collision-dependent and took place with both the *XBP1u* and the poly(A) endogenous stallers.

Therefore, we decided to structurally characterize human ribosomes collided on the *XBP1u* mRNA as an endogenous substrate for ZNF598-mediated ubiquitination. Interestingly, we observed two additional inter-ribosomal contact sites, which were not mentioned for the structure of the rabbit disome[10] (Fig. 2). One of these contact sites bridges the large subunit of the colliding ribosome and the small subunit of the leading ribosome, which was so far only described in yeast disomes[9]. The second additional contact takes place between the 18S rRNA expansion segment ES6c of the colliding ribosome (ES6c-2) and the eS15 and eS27 proteins of the leading ribosome (eS15-1, eS27-1; Fig. 2). We, therefore, propose that there are four main inter-ribosomal contact sites in the mammalian disome which can represent specific binding interfaces for collision sensor factors such as ZNF598. Nonetheless, the human and rabbit collided disomes are largely equivalent in terms of the general architecture and we used the easily accessible collided ribosomes from the rabbit translation system for further experiments. On the other hand, we observed striking differences when comparing the structure of human and yeast collided disomes. The position of the colliding ribosome in the yeast disome seems rotated by 18° around an axis longitudinal to its 40S subunit respective to the human colliding ribosome. This apparent rotation could explain why the yeast eS10 does not seem to be the primary ubiquitination target of Hel2[1]. While the structured part of the yeast eS10 is approximately 10–12 Å farther away from the stalled ribosome than the human eS10, it is hard to judge how this affects the ubiquitination site. The C-terminal part of eS10 harboring the two target lysine residues ubiquitinated by Hel2/ZNF598 is unstructured and not present in the structural model. Overall, both ribosomes appear to be in closer contact with yeast, which may explain the observed higher stability and smaller hinging movements in the yeast disome. Consequently, the increased stability could contribute to easier recognition and more efficient subunit splitting for RQC initiation.

To study the ubiquitination of collided mammalian ribosomes, we reconstituted ZNF598-mediated polyubiquitination using endogenous

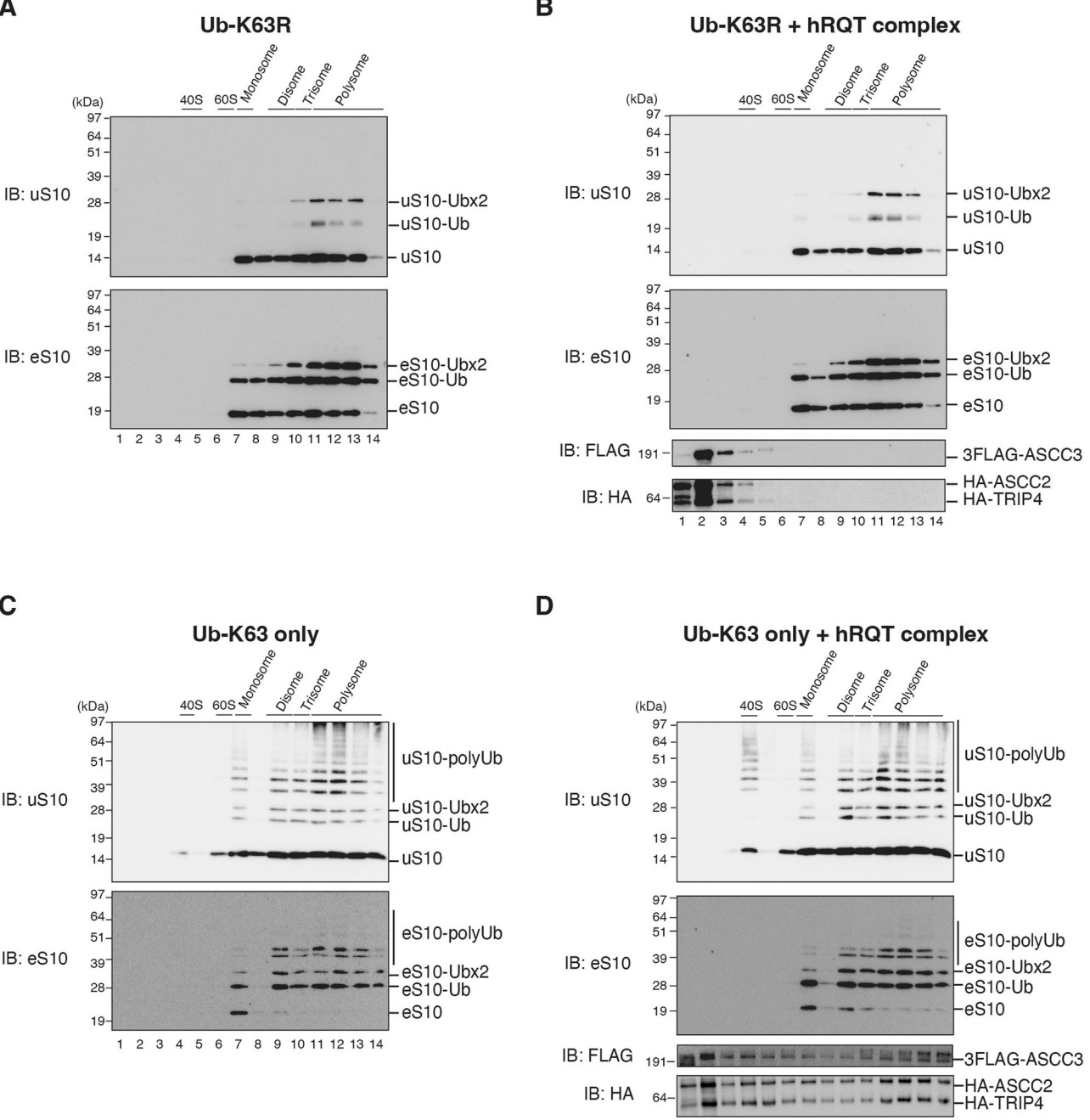

**Fig. 6 | The hRQT complex dissociates collided ribosomes with K63-linked polyubiquitin chains on uS10. A–D** In vitro hRQT complex-mediated subunit disassociation assay of K63-linked polyubiquitinated ribosomes. Isolated RNCs generated by *XBP1u* staller mRNA were ubiquitinated by ZNF598 with either Ub-K63R (**A**, **B**) or Ub-K63only (**C**, **D**). The ubiquitinated RNCs were incubated with (**B**, **D**) or without (**A**, **C**) the hRQT complex. After hRQT-mediated splitting reactions, samples were subjected to ultracentrifugation through sucrose density gradients and gradient fractions were analyzed by western blotting using indicated antibodies. We obtained essentially the same results in at least three independent experiments.

*XBP1u* and poly(A) stalling sequences. ZNF598-mediated reaction with specific ubiquitin mutants, the Ub-K63R or the Ub-K63only, revealed that ZNF598 formed mainly K63-linked polyubiquitin chains on uS10 and eS10 of the purified collided ribosomes (Figs. 5C, 6C). The subsequent hRQT complex-mediated subunit dissociation reaction produced 40S subunits with detectable polyubiquitinated uS10 but not eS10 (Fig. 4D, F). This suggests that the hRQT complex recognizes polyubiquitinated uS10. Moreover, the dissociation of the ubiquitinated 40S was only observed with K63-linked polyubiquitinated uS10 (Fig. 6D). When K63-linked polyubiquitination was not possible (using the K63R ubiquitin mutant), no subunit dissociation by hRQT was

observed (Fig. 6B). Together, these results show that only collided ribosomes with K63-linked polyubiquitinated uS10 are subjected to the hRQT complex-mediated subunit dissociation. This is a striking result because, in agreement with previously published studies[4–6], we also see that the majority of eS10 is ubiquitinated while the uS10 ubiquitination seems minor in comparison (Figs. 4C–F, 5C, F, G, 6 and S6). Interestingly, we observed the background of monoubiquitinated eS10 even without anisomycin treatment in HEK293T cells (Supplementary Fig. 1C). This background corresponds to the one observed in many of the in vitro experiments (such as in Figs. 1, 4–6) and points at a general propensity of eS10 to be ubiquitinated. As mentioned above, the minor

uS10 polyubiquitination is readily detected in 40S subunits dissociated by hRQT, which is not the case for ubiquitinated eS10. Here again, the minimal eS10 monoubiquitination was observed, likely originating from dissociation of collided ribosomes with polyubiquitinated uS10. It is also possible that eS10 of the leading stalled ribosome as the primary target of hRQT is not ubiquitinated. Alternatively, this could suggest a different role of eS10 ubiquitination in collided ribosomes.

Notably, we also detected some ZNF598-mediated ubiquitination in the monosome fractions (Figs. 4C, E, 5F, G and 6C). This has been reported before[3,10] and there are three plausible explanations. First, the ubiquitinated monosome product of the RQT-mediated disome dissociation cannot be further dissociated by the RQT complex[27]. Second, the free 60S and the ubiquitinated 40S products of RQT-mediated subunit dissociation can re-associate. Finally, ubiquitinated 80S ribosomes can dissociate from polysomes during the purification and incubation steps.

We observed the ZNF598-mediated polyubiquitination and the hRQT-mediated subunit dissociation dependent on K63-linked ubiquitination using ribosomes collided on the *XBP1u* staller mRNA (Fig. 6). This demonstrates the conserved mechanism of RQC induced by endogenous stalling sequences and confirms that the K63-linked polyubiquitination of collided ribosomes is crucial for the hRQT-mediated subunit dissociation. However, it was not clear how individual components of the hRQT/ASC-1 complex contribute to its activity in the cytoplasm. It has been demonstrated that the ASC-1 complex dissociates collided ribosomes into subunits[8]. In this study, the hRQT complex composed of ASCC3, ASCC2, and TRIP4 was sufficient to also dissociate collided ribosomes. This suggests that the ASCC1 subunit of the nuclear ASC-1 complex is dispensable for its ribosome splitting activity in the cytoplasm as previously proposed[7,8]. We also confirmed that the ATPase activity of ASCC3 is required for the hRQT complex-mediated subunit dissociation. Furthermore, we studied the controversial role of ASCC2 ubiquitin-binding activity of the hRQT complex. We used mutations in the ubiquitin-binding domain of ASCC2, which disrupted the subunit dissociation activity of hRQT (Fig. 5F). Moreover, we observed that ASCC2 is directly associated with the K63-linked polyubiquitin chains (Fig. 6D). Together, these results show that the K63-linked polyubiquitin chain binding capacity of ASCC2 is required for the hRQT-mediated reaction (Fig. 5F).

Recent studies revealed crucial roles of ribosomal collisions in the inhibition of translation initiation[28,29], induction of integrated stress response[30–32], apoptosis[20], and innate immune response[33]. It is crucial to understand how key regulators recognize ribosomal collisions, since the hRQT antagonizes these signaling pathways by dissociating the colliding ribosomes. In this regard, the substrate specificity of the hRQT and the particular ubiquitination mode of human ribosomes collided on endogenous stallers play a key role.

## Methods

### Plasmid construction
All recombinant DNA techniques were performed according to standard procedures using *E. coli* DH5α cells for cloning and plasmid propagation. Plasmids used in this study are listed in Supplementary Table 2. DNA cloning was performed by PCR amplification with gene-specific primers using PrimeSTAR HS DNA polymerase (#R010A, Takara-bio, Shiga, Japan) and T4 DNA ligase (#M0202S, NEB, Ipswich, MA 01938-2732, USA). CSII-CMV-MCS-IRES2-Bsd (RDB04385), pCAG-HIVgp (RDB04394), and pCMV-VSV-G-RSV-Rev (RDB04393) were kindly provided by Dr. Hiroyuki Miyoshi (RIKEN BioResource Center, Ibaraki, Japan)[34]. Human ribosomal protein cDNAs were amplified by PCR using KOD FX Neo (TOYOBO, KFX-201) from HEK293T cDNA. The cDNAs were inserted into the pCMV-HA-C vector or pCMV-FLAG-C (TAKARA, 635690 or 635688). 3xHA-tag or 3xFLAG tag sequences were included in PCR primers. The ribosomal protein mutants were generated using KOD mutagenesis Kits (TOYOBO, SMK-101). Lentiviral vectors were constructed by subcloning ribosomal protein cDNAs into CSII-CMV-MCS-IRES2-Bsd. All cloned DNAs amplified by PCR were verified by sequencing.

### Cell culture, transfection
HEK293T (RCB2202) cells were provided by the RIKEN BRC through the National Bio-Resource Project of MEXT, Japan. They were grown in Dulbecco's modified Eagle's medium (DMEM) with 10% fetal bovine serum (FBS) and penicillin/streptomycin (PS) (100 U/mL). 293FT cells (R700-07) were from Thermo Fisher and were grown in DMEM with 10% FBS, 1x Non-Essential Amino Acids Solution (Thermo Fisher 11140-050), and PS (100 U/mL). Plasmid transfection was performed using the PEI-MAX reagent (Cosmo Bio, Koto, Tokyo, Japan). Used cell lines in this study are listed in Supplementary Table 3.

### Lentivirus production and infection
293FT cells were grown to around 80% confluency on 6-cm plates. Transfection mixture [14.7 μL Lipofectamine 2000 (Thermo Fisher, 11668-019), 1.2 μg pCAG-HIVgp (RDB04394), 1.2 μg pCMV-VSV-G-RSV-Rev (RDB04393), and 2.5 μg CSII-CMV-MCS-IRES2-Bsd (RDB04385) containing each ribosomal protein cDNA] was prepared in 500 μL Opti-MEM reduced serum medium (Thermo Fisher, 31985-062). The mixture was incubated at room temperature for 15 min and gently added to the 293FT cells for 24 h incubation (37 °C, 5% CO$_2$). The medium was then replaced with pre-warmed media (DMEM supplemented with 10% FBS). Forty-eight hours after the start of the transfection, lentivirus-containing cell culture supernatants were collected and filtered (Millipore, MILLEX GV 0.45 μm). The resultant virus solution (4.5 mL) supplemented with polybrene (Sigma, H9268, 5 μg/mL) was added to HEK293T cells that were seeded at $4 \times 10^5$ cells per 6-cm dish the day before infection. Forty-eight hours after infection, one-fifth of the infected cells were seeded on new 10-cm plates following trypsinization and further incubated in the presence of blasticidin-S (WAKO, 029-18701, 10 μg/mL) for 48 h to select infected cell populations.

### Preparation of cell lysate
Cells were treated with anisomycin at the indicated concentrations for 15 min (37 °C, 5% CO$_2$). Cells were washed with PBS twice and lysed with RIPA buffer (20 mM Tris-HCl [pH 7.5], 150 mM NaCl, 2 mM EDTA, 10 mM sodium fluoride, 10 mM β-glycerophosphate, 1% Triton X-100, 0.1% SDS, 0.5% sodium deoxycholate, 40 mM *N*-Ethylmaleimide). Cell lysates (15 μg) were analyzed by immunoblot.

### Electrophoresis and western blotting
For neutral PAGE, cells were lysed with passive lysis buffer (Promega) and centrifuged at $13500 \times g$ for 1 min. Supernatants were collected, and equal amounts of total proteins were used as protein samples. For RNase(+) samples, RNase A (QIAGEN) was added at a final concentration of 0.05 mg/ml and incubated on ice for 20 min; for RNase(−) samples, Milli-Q water was added instead. After incubation, 1x Sample Buffer (200 mM Tris pH 6.8, 8% w/v SDS, 40% glycerol, 0.04% BPB, and 100 mM DTT) was added and heated at 65 °C for 10 min. Proteins were separated by 15% PAGE under neutral pH conditions (pH 6.8) for 4 h with a 150 V constant voltage in MES-SDS buffer (1 M MES, 1 M Tris base, 69.3 mM SDS, and 20.5 mM EDTA) and were transferred to a PVDF membrane (#IPVH00010, Millipore). Protein samples for in vitro translation experiment were prepared with iced cold 100% Tri-chloroacetic acid, dissolved with 1 x Sample Buffer, and analysed as previously described in ref. 7. Used primary and secondary antibodies for western blotting in this study are listed in Supplementary Table 4. For CBB staining, proteins were separated and stained with homemade CBB solution (0.1% Coomassie Brilliant Blue R-250, 45% MeOH, 10% acetic acid) and washed with homemade de-staining solution (25% isopropanol and 10% acetic acid).

## Purification of recombinant proteins

To obtain ZNF598 from Lenti-X 293 T cells, Lenti-X 293 T cells over-expressing 3FLAG tagged ZNF598 under the control of the CMV promoter were grown exponentially in one 10 cm dish at 37 °C. After washing with ice-cold PBS, cells were lysed by passage through a 25 G syringe in ice-cold buffer 1 (10 mM HEPES pH 7.4, 500 mM NaCl, 10% Glycerol, 2 mM DTT, 2% NP-40, 1 mM PMSF) containing cOmplete Mini EDTA-free Protease Inhibitor Cocktail (#11836170001, Roche, Basel, Basel-Stadt, Switzerland) (1 tablet/10 mL) and then centrifuged at 7300 × $g$, at 4 °C for 10 min for three times to obtain a clear lysate. To purify FLAG-tagged proteins, pre-equilibrated anti-DYKDDDDK tag antibody beads (#016−22784, Fujifilm Wako pure chemical corporation, Osaka, Osaka, Japan) were incubated at 4 °C for 2 h in the obtained lysate, washed with buffer 1 for seven times, and then eluted with 500 μL of buffer 2 (10 mM HEPES pH 7.4, 100 mM NaCl, 10% Glycerol, 2 mM DTT, and 1 mM PMSF) containing 250 μg/mL FLAG peptide (GenScript Biotech, Piscataway, NJ, USA) for 3 h.

To obtain hRQT complex from Lenti-X 293 T cells, Lenti-X 293 T cells overexpressing indicated RQT factors containing 3FLAG tagged ASCC3 under the control of the CMV promoter were grown exponentially in three 15 cm dishes at 37 °C. After washing with ice-cold PBS, cells were lysed by passage through a 25 G syringe in ice-cold buffer 3 (50 mM Tris pH 7.5, 100 mM NaCl, 10 mM MgCl$_2$, 10% Glycerol, 5 mM 2-mercaptoethanol, 1% NP-40, 10 μM ZnCl$_2$, 100 mM L-arginine, 1 mM PMSF) containing cOmplete Mini EDTA-free Protease Inhibitor Cocktail and then centrifuged at 7300×$g$, at 4 °C for 10 min for three times to obtain a clear lysate. To purify the FLAG-tagged complex, pre-equilibrated anti-DYKDDDDK tag antibody beads were incubated at 4 °C for 1.5 h in the obtained lysate, washed with buffer 4 (50 mM Tris pH 7.5, 300 mM NaCl, 10 mM MgCl$_2$, 10% Glycerol, 5 mM 2-mercaptoethanol, 1% NP-40, 10 μM ZnCl$_2$, 100 mM L-arginine, and 1 mM PMSF) for five times, and then eluted with 400 μL of buffer 4 containing 250 μg/mL FLAG peptide (GenScript Biotech, Piscataway, NJ, USA) for 2 h.

To obtain Rqc2-FLAG from *S. cerevisiae*, Rqc2-FLAG was purified from 1 L of SC 2% glucose culture of yeast cell harboring p*GPDp-Rqc2-FLAG-TEV-ProteinA-CYC1t*. The harvested cell pellet was frozen in liquid nitrogen and then ground in liquid nitrogen using a mortar. The cell powder was resuspended with lysis buffer 500 (50 mM HEPES pH 7.5, 500 mM KOAc, 2.5 mM Mg(OAc)$_2$, 10 μM ZnCl$_2$, 0.01% NP-40, 10% Glycerol, 1 mM DTT) containing 1 pill/10 ml of complete-mini EDTA free, to prepare the lysate. The lysate was centrifuged at 39,000 × $g$ for 30 min at 4 °C, and the supernatant fraction was used for the purification step. Rqc2-FTP was affinity purified using IgG beads (GE Healthcare), then cleaved with TEV protease to release it and repurified using anti-DYKDDDDK tag antibody beads (WAKO) as previously described in ref. 3.

To obtain Ubc13 and Mms2 from *E.coli*, the ubiquitin-conjugating enzymes Ubc13 and Mms2 were cloned into a pGEX6P1 vector and purified as tag-free recombinant proteins. The expression of GST-Ubc13 and GST-Mms2 was induced by the addition of IPTG at 30 °C for 6 h. For purification of tag-free proteins, these were eluted by Pre-Scission protease (# 27084301, GE Healthcare) from Glutathione Sepharose 4B (# 17-1756-05, GE Healthcare) as previously described in ref. 35.

## Ubiquitin-binding assay

To produce the K63-linked polyubiquitin chains, 5 μM Ubiquitin (#E1100, UBPBio), 50 nM UBE1 (# B1100, UBPBio), 200 nM Ubc13, 200 nM Mms2 and energy regenerating source (1 mM ATP (Roche), 10 mM creatine phosphate (# 030-04584, Wako), and 20 μg/ml creatine kinase (# 10127566001, Roche)) were mixed in buffer 5 (50 mM Tris-HCl pH 7.5, 100 mM NaCl, 10 mM MgCl2, 1 mM DTT, 10% Glycerol) at 32 °C for 3.5 h. The K63-linked polyubiquitin chain product or purchased ubiquitin (#D2300, UBPBio) were incubated with the purified

hRQT complex-conjugating anti-FLAG® M2 Magnetic Beads (#M8823-5ML, SIGMA) in buffer 6 (50 mM Tris pH 7.5, 100 mM NaCl, 2.5 mM MgCl$_2$, 10 μM ZnCl$_2$, 10% Glycerol, 1 mM DTT, 0.1% NP-40, 100 mM L-arginine, 1 mM PMSF) at 23 °C for 15 min. To prevent non-specific binding, beads were washed five times and then eluted by 1 x Sample Buffer. In the analysis using purchased tetraubiquitin chains (Ubiquitin #E1100, K48-Ub4 #D1300, K63-Ub4 #D2300, UBPBio), 1.5 μg of mono- or K48-tetraubiquitin, 125 ng of K63-tetraubiquitin were incubated with the purified hRQT complex-conjugating anti-FLAG® M2 Magnetic Beads as described above.

## Sucrose density gradient ultracentrifugation

Sucrose density gradients (15–45% sucrose in 50 mM HEPES pH 7.4, 100 mM KOAc, 5 mM Mg(OAc)$_2$) were prepared using a Gradient Master (BioComp). The purified RNCs were applied on a sucrose gradient, and ribosomal fractions were separated by centrifugation for 1.5 h at 201,000×$g$ (avg.) at 4 °C in a P40ST rotor. The polysome profiles were generated by continuous absorbance measurement at 260 nm using a single path UV-1 optical unit (ATTO Biomini UV-monitor) connected to a chart recorder (ATTO digital mini-recorder).

## mRNA preparation

*Xbp1u* reporter mRNA was produced using the corresponding PCR fragment and mMESSAGE mMACHINE T7 transcription Kit (Thermo Fisher, Waltham, MA, USA) according to the manufacturer protocol. PCR fragment was amplified using pTX573 as a DNA template and the indicated primers listed in Supplementary Table 5. Generated 5′ Capped *Xbp1u* mRNAs were designed to be T7 promoter, Kozak sequence, initiator AUG codon, His6-tag for RNC purification, PA-tag for immunoblotting, and coding sequence of *XBP1u* (from 5′ end). After in vitro transcription reaction and TURBO DNase treatment, mRNAs were purified using RNeasy Mini Kit (QIAGEN).

## In vitro translation and RNC purification

All in vitro translation analysis were performed using Rabbit Reticulocyte Lysate System, Nuclease Treated (Promega). Translation with generated mRNA was at 30 °C for 15 min unless indicated otherwise in the Fig. legend. The stalled RNCs on the *Xbp1u* mRNA were affinity purified using the His6-tag on the nascent polypeptide chain and magnetic beads. After the translation reaction, RRL was adjusted to 750 mM KOAc, 25 mM Mg(OAc)$_2$, 250 mM Sucrose, and 2 mM Spermidine, and transferred on ice to stop the reaction. The lysate was applied to Dynabeads for His6-tag isolation and pulldown (Invitrogen) and buffer 7 (50 mM HEPES pH 7.4, 750 mM KOAc, 25 mM Mg(OAc)$_2$, 250 mM Sucrose, 5 mM 2-mercaptoethanol, 0.01% NP-40, 2 mM Spermidine, RNasein Plus RNase Inhibitor (Promega)) at 4 °C for 10 min. After washing by buffer 7 for five times, tethered RNCs were eluted with buffer 8 (50 mM HEPES pH 7.4, 100 mM KOAc, 25 mM Mg(OAc)$_2$, 250 mM Sucrose, 5 mM 2-mercaptoethanol, 0.01% NP-40, 2 mM Spermidine, RNasein Plus RNase Inhibitor) containing 300 mM Imidazole at 20 °C for 15 min.

## In vitro ubiquitination of stalled RNCs

Ubiquitination coupled with translation in vitro. For the reconstitution of translational pausing-coupled RNC ubiquitination, 100 nM of UBE1, 250 nM UBE2D3 (Funakoshi), and 10 nM of purified 3FLAG-ZNF598 protein were supplemented in the RRL during in vitro translation reaction at 20 °C for 60 min. After the translation reaction, the same handling was performed as described above.

Ubiquitination of RNC tethered on beads. For depleting endogenous ubiquitin and using exogenous ubiquitin mutants, we performed a ubiquitination reaction on RNCs-tethered magnetic beads. After in vitro translation with 200 μl of RRL and binding purified RNCs to magnetic beads, RNCs-beads were washed with buffer 7 for three times and buffer 8 for two times, then resuspended with 100 μl of

buffer 8 by gently pipetting. To pre-charge the E2 with ubiquitin, 5 µl of 500 µM tag-free ubiquitin wild-type (# E1100, UBPBio) or mutants as described below, 1 µl of 15 µM UBE1, 6 µl of 25 µM UBE2D3, and 15 µl of 10x ERS (10 mM ATP, 100 mM creatine phosphate, 200 µg/ml creatine kinase) were mixed and incubated at 23 °C for 15 min. Then, 100 µl of resuspended RNCs-beads, pre-charged ubiquitination components, 1 µl of RNasein Plus RNase Inhibitor, and 100 nM of purified 3FLAG-ZNF598 protein were mixed, adjusted to 250 mM Sucrose, and reacted at 20 °C with rotating for 30 min. In other words, the final reaction composition is (30 µM Tag-free ubiquitin, 100 nM UBE1, 200 nM UBE2D3, 1 mM ATP, 10 mM creatinine phosphate, 20 µg/ml creatinine kinase, 100 nM 3FLAG-ZNF598, RNCs derived from 200 µl of RRL, 250 mM Sucrose, RNasein Plus RNase Inhibitor in buffer 8). After the ubiquitination reaction, the same handling was performed as described above. The following tag-free ubiquitin mutant proteins were purchased from Boston Biochem: K63only (#UM-K63O), K63R (#UM-K63R), and K0 (#UM-NOK).

Ubiquitination of isolated colliding RNCs. Translation lysate was separated by the ultracentrifugation before the ubiquitination reaction. After RNC purification followed by fractionation with sucrose density gradient centrifugation, collected polysome fractions were mixed and incubated with the ubiquitination mixture (10 µM His6-tagged ubiquitin wild-type or mutants, 50 nM UBE1, 200 nM UBE2D3, 1 mM ATP, 10 mM creatinine phosphate, 20 µg/ml creatinine kinase, and 5 nM 3FLAG-ZNF598 in buffer 8) at 30 °C, and harvested at the indicated time point. The following His6-tagged ubiquitin proteins were purchased from UBPBio: WT (#E1300), K63only (#E1880), K63R (#E1780), and K0 (#E1710).

### In vitro splitting reaction of ubiquitinated and collided RNCs

After the ubiquitination reaction of RNCs on magnetic beads, RNCs-beads were washed with buffer 8 for one time, buffer 7 for two times and buffer 9 (50 mM HEPES pH 7.4, 300 mM KOAc, 5 mM Mg(OAc)$_2$, 250 mM Sucrose, 5 mM 2-mercaptoethanol, 0.01% NP-40, 2 mM Spermidine, RNasein Plus RNase Inhibitor) for two times, and then eluted with buffer 9 containing 300 mM Imidazole at 20 °C for 15 min. For the splitting reaction, eluted RNCs were incubated with 5 nM of purified hRQT complex, 100 nM of purified Rqc2 as a reassociation inhibitor, and 1 mM ATP in buffer 9 at 20 °C for 45 min. After incubation, ribosomal fractions were separated via sucrose density gradient centrifugation. The RNCs were monitored via UV absorbance at a 260 nm wavelength and detected with immunoblotting.

### Quantification and statistical analysis

All blot experiments were repeated three times independently.

### Cryo-EM sample preparation of collided human disomes

*XBP1-XTEN* reporter mRNA was produced using the corresponding PCR fragment and mMESSAGE mMACHINE T7 transcription Kit (Thermo Fisher, Waltham, MA, USA) according to the manufacturer protocol as described above. PCR fragment was amplified using the *XBP1-XTEN* pUC57 plasmid as a DNA template and the indicated primers listed in Supplementary Table 4. Generated 5' Capped *XBP1-XTEN* (Supplementary Fig. 1A) mRNA was used in an in vitro translation reaction utilizing translation extract from HeLa S3 cells. HeLa S3 translation extract was prepared essentially as described before[36]. In brief, HeLa S3 cells were grown to a density of $3.0–5.5 \times 10^5$ in SMEM (Sigma), supplemented with 10% heat-inactivated FBS (Gibco), Penicillin (100 U/ml)/Streptomycin 100 µg/ml (Gibco) and 1x GlutaMAX (Gibco) at 37 °C, 5% CO$_2$ using a spinner flask (90 rpm). Cells were treated with 200 nM integrated stress response inhibitor (ISRIB) 1 h prior to harvesting to ensure cap-dependent translation initiation[37]. Subsequently, cells were harvested (2 min, 650×*g*) and the resulting pellet was washed 3x with Washing Buffer (35 mM HEPES/KOH pH 7.5, 140 mM NaCl, 11 mM Glucose) (1 min, 650 × *g*) and 1x with Extraction

Buffer (20 mM HEPES/KOH pH 7.5/4 °C, 45 mM KOAc, 45 mM KCl, 1.8 mM Mg(OAc)$_2$, 1 mM DTT). Further, the cell pellet was resuspended in Extraction Buffer ($1.2 \times 10^9$ cells/ml) and disrupted by nitrogen pressure (300 psi, 30 min, 4 °C) in a cell disruption vessel (Parr Instrument). The lysate was mixed with 1/29 volume High Potassium Buffer (20 mM HEPES/KOH pH 7.5/4 °C, 945 mM KOAc, 945 mM KCl, 1.8 mM Mg(OAc)$_2$, 1 mM DTT), incubated at 4 °C for 5 min and cleared by centrifugation (15 min, 20,817×*g*, 4 °C). Aliquots of the resulting supernatant were frozen in liquid nitrogen and stored at −80 °C.

For the in vitro translation reaction, 3 mL translation reaction mix with 50% (v/v) extract was adjusted to 2.75 mM Mg(OAc)2, 0.42 mM MgCl2, 75 mM KOAc, 37.5 mM KCl, 42 mM NaCl, 2 mM DTT, 1.56 mM GTP, 0.25 mM ATP, 1.6 mM creatine phosphate, 0.45 mg/mL creatine kinase, 50 µg/mL yeast tRNA, 0.4 mM spermidine, 0.12 mM complete amino acid mixture (Promega), and 0.8 U/µL RNase inhibitor (Invitrogen). The reaction was then initiated by the addition of 90 µg of *Xbp1-XTEN* mRNA and incubated for 70 min at 17 °C. Subsequently, ribosomes were isolated from the reaction by pelleting through a sucrose cushion (50 mM HEPES-KOH pH 7.5, 10 mM Mg(OAc)$_2$, 200 mM KOAc, 1 mM DTT, 0.01% NP-40, and 1.5 M sucrose) in a TLA100 rotor for 1 h at 434,513 × *g* and 4 °C. The ribosome pellet was resuspended in purification buffer (50 mM HEPES pH 7.5, 100 mM KOAc, 10 mM Mg(OAc)$_2$, 50 mM sucrose, 5 mM 2-mercaptoethanol, 0.1% Nikkol, and 1 mM spermidine) and RNCs were purified using the His-tag of the nascent polypeptide chain encoded by the *Xbp1-XTEN* mRNA and magnetic Dynabeads His-Tag Isolation and Pulldown beads (Invitrogen). The affinity purification was done essentially as described before[38]. In brief, the in vitro translation reaction was added to the beads and incubated while rotating for 15 min at 4 °C. The beads were washed three times with excess of a wash buffer (50 mM HEPES/KOH, pH 7.5, 100 mM KOAc, 25 mM Mg (OAc)$_2$, 250 mM sucrose, 0.1% Nikkol, and 5 mM ß-Mercaptoethanol) and eluted in 400 µl of the same buffer containing 350 mM imidazole. The elution was applied to a 10–50% sucrose gradient in wash buffer, and ribosomal species were separated by centrifugation for 3 h at 172,000 × *g* at 4 °C in an SW40 rotor. For gradient fractionation, a Piston Gradient Fractionator (BIO-COMP) was used. The disome fraction was collected and pelleted over 400 µl sucrose cushion buffer at 534,000 × *g* for 45 min at 4 °C in a TLA110 rotor. The resulting ribosomal pellets were resuspended carefully on ice in 25 µL of grid buffer (20 mM HEPES/KOH, pH 7.2, 50 mM KOAc, 5 mM Mg(OAc)$_2$, 125 mM sucrose, 0.05% Nikkol, 1 mM DTT, and 0.01 U/µl SUPERase-IN (Invitrogen).

### Cryo-EM data acquisition and processing

A freshly prepared sample of the concentrated disome fraction was applied to 2 nm carbon pre-coated Quantinfoil R3/3 holey carbon support grids and vitrified using Vitrobot Mark IV (Thermo Fisher). Data were collected at Titan Krios TEM (Thermo Fisher) equipped with a Gatan K2 Summit direct detector at 300 keV under ~43.6 e-/Å2 over 40 frames in total, and a defocus range of −0.4 to −3.5 µm using the EPU 2.12.1 software. Magnification settings resulted in a pixel size of 1.045 Å per pixel. Original image stacks were summed and corrected for drift and beam-induced motion at the micrograph level by using MotionCor2 1.4.0[39] The contrast transfer function (CTF) estimation and resolution range of each micrograph were performed with Gctf 1.06[40]. All cryo-EM data were processed using standard procedures with GAUTOMATCH 0.56 (http://www.mrc-lmb.cam.ac.uk/kzhang/) used for particle picking and cryoSPARC 3.2[41] and Relion 3.1.3[42] for 2D and 3D classification, 3D reconstructions, and further processing.

Two datasets were collected and initially processed separately (Supplementary Fig. 2). To avoid any initial bias, the datasets were processed using an 80S extension approach as reported previously[3,9]. In brief, individual 80S particles were picked and subjected to 2D classification. A total of 1,115,769 particles were selected after 2D classification. Initial refinements and 3D classifications into six classes

were performed for each dataset. Two classes representing the leading and the colliding 80S within the disome showing the strongest density for their respective collision partner ribosome were selected for further classification and processing.

We focused on classes of the last ribosome in the A/P P/E state for further processing as these showed the most prominent density of the collision partner at the mRNA entry side and combined these particles from both datasets. These particles were stepwise re-extracted while expanding the box size, centered, and 3D refined, allowing for large shifts with the final expansion of the box size to 900 pixels (downsampled to 300 pixels). Local refinement with signal subtraction of individual ribosomes from this downsampled consensus volume yielded the first stalled ribosome in the non-rotated P/P E/E state analogous to our previous structures[9,12]. Corresponding individual ribosomes were resolved to an overall resolution of 2.9 (first stalled) and 3.2 Å (second colliding) and fitted into the consensus refinement to create a composite cryo-EM density map with an overall resolution of 3 Å (Supplementary Fig. 3). Resolution of the composite map was estimated using composite half maps and the gold-standard Fourier shell correlation criterion (FSC = 0.143; 2.96 Å) in cryoSPARC and the map-model FSC (FSC = 0.5; 2.98 Å) in Phenix 1.19[43].

### Model building
To generate a molecular model for our structure, we used previously published models of human 80S ribosomes in hybrid (PDB ID: 6Y57) and post (PDB ID: 6Y2L) states[44] and a model of human E/E tRNA (PDB ID: 6Z6L)[45]. The density of the 18 S rRNA expansion segment ES6c of the colliding ribosome becomes prominent only at a lower contour level and given the low local resolution, we could not build a de novo molecular model for ES6c with sufficient confidence. Therefore, we used the 3dRNA[46] structure predicted for the ES6c sequence (residues 692–738), which we could fit into the density (Supplementary Fig. 3D). First, individual ribosomes, tRNA and the expansion segment were fitted as rigid bodies into the densities. These models were then remodeled and refined in COOT 0.9[47] and Phenix 1.19[43]. Cryo-EM densities and models were visualized in UCSF ChimeraX 1.3[48]. Detailed statistics of model refinements and validation are listed in Supplementary Table 1.

### Reporting summary
Further information on research design is available in the Nature Research Reporting Summary linked to this article.

## Data availability
The cryo-EM structural data generated in this study have been deposited in the Protein Data Bank and in the Electron Microscopy Data Bank databases under accession codes PDB: 7QVP and EMDB-14181. Source data are provided as a Source Data file.

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

## Acknowledgements

This study was supported by AMED (grant 20gm1110010h0002 to T.I.) and MEXT/JSPS KAKENHI under Grant Numbers 19H05281, 21H05277, 22H00401 (to T.I.), 21H00267 and 21H05710 (to Y.M.). This study was also supported by Takeda Science Foundation (to T.I.) and by JST PREST Grant Number JPMJPR21EE to Y.M. and by the German Research Council (BE1814/15-1) to R.B. T.D. is supported by the Graduate School of Quantitative Biosciences Munich (QBM).

## Author contributions

The in vitro reconstitution experiments were designed and interpreted by M.N., T.Sugiyama, Y.M., and T.I.; M.N. and T.Sugiyama performed the experiments under the supervision of T.I. and Y.M.; The experiment to monitor the ribosome collision induced by anisomycin were performed by C.K., S.I, N.S., M.N., and T.Suzuki; S.H. constructed the cell-line to purify the hRQT complex under the supervision of T.I.; Cryo-EM sample was prepared by P.T. and I.M., data were processed by T.D., and molecular model was built and refined by T.D and P.T. Structural data were analysed and interpreted by T.D. and P.T. under the supervision of R.B. T.I. and R.B. primarily conceived the idea and designed the experiments. T.I., M.N., P.T., T.Suzuki, T.Sugiyama, T.D., and R.B. wrote the manuscript. T.I. supervised the project.

## Competing interests

The authors declare no competing interests.

## Additional information

# Article

**Momoko Narita**[1,2,5], **Timo Denk**[3,5], **Yoshitaka Matsuo** ⓘ[1], **Takato Sugiyama**[2], **Chisato Kikuguchi**[1], **Sota Ito**[1], **Nichika Sato**[1], **Toru Suzuki**[1], **Satoshi Hashimoto** ⓘ[2], **Iva Machová**[4], **Petr Tesina**[3], **Roland Beckmann** ⓘ[3,6] ✉ & **Toshifumi Inada** ⓘ[1,2,6] ✉

[1]Division of RNA and gene regulation, Institute of Medical Science, The University of Tokyo, Minato-Ku 108-8639, Japan. [2]Graduate School of Pharmaceutical Sciences, Tohoku University, Sendai 980-8578, Japan. [3]Gene Center and Department of Biochemistry, University of Munich, Feodor-Lynen-Str. 25, 81377 Munich, Germany. [4]Biomedical Centre, Faculty of Medicine in Pilsen, Charles University in Prague, Alej Svobody 1655/76, 323 00 Pilsen, Czech Republic. [5]These authors contributed equally: Momoko Narita, Timo Denk. [6]These authors jointly supervised this work: Roland Beckmann and Toshifumi Inada. ✉e-mail: beckmann@genzentrum.lmu.de; toshiinada@ims.u-tokyo.ac.jp

