## [Peer Review File · Nature Communications]

A distinct mammalian disome collision interface harbors K63-linked polyubiquitination of uS10 to trigger hRQT-mediated subunit dissociationREVIEWER COMMENTS

Reviewer #1 (Remarks to the Author):

Ribosome collisions are recognized by the ribosome-associated quality control system and important for rescuing stalled ribosomes in all domains of life. In yeast, Hel2 recognizes collided ribosomes and polyubiquitinates uS10 on the collided ribosome, a mark that is recognized by the RQT complex to promote subunit dissociation. In mammals, ZNF598-mediated ubiquitination of both uS10 and eS10 has been reported, raising the question as to what exactly the signal is that is recognized for ribosome splitting. Here the authors demonstrate that while ZNF598 indeed ubiquitinates both uS10 and eS10, only the polyubiquitination of uS10 (and not monoubiquitination of uS10 or eS10) is recognized by the human RQT complex to mediate subunit splitting. Moreover, the authors demonstrate that subunit dissociation requires K63-linked polyubiquitination of uS10 as well as the ATPase activity of ASCC3 of the hRQT complex. These latter points represent the strength of the paper and the biochemical experiments are technically excellent, nicely presented and appropriately interpreted. The authors also present a cryo-EM structure of a human collided ribosome that appears similar to rabbit, but distinct from yeast. It is a little disappointing that it is not the modified collided ribosome or a complex with ZNF598 or RQT. It is also quite confusing in the way that the authors highlight differences with the rabbit system, yet all the biochemistry indicates that these systems are analogous. Nevertheless, overall, I believe it is an important piece of work that will be of interest to the scientific community, especially those working on quality control, however, it could benefit from addressing a few mostly minor points.

1. Lines 118-119 states that uS10 and eS10 ubiquitination by ZNF598 have been reported previously (reference 4)...however in line 125, it is written as though the detection of polyubiquitination of uS10 is a new finding and makes the mammalian system now analogous with yeast. Presumably the distinction relates to mono- versus poly-, but this is not made clear. In the introduction, the authors write on line 71 that ZNF598 was proposed to preferentially monoubiquitinate eS10 and uS10 in mammals (references 4-6). Given that the whole paper revolves around these modifications maybe some more details need to be provided in the introduction as to why it is just a proposal as opposed to an observation supported by experiments? There are two aspects related to which protein eS10 and uS10 as well as mono- versus polyUb...but these are all jumbled together and it is not explained clearly enough for a non-expert in the introduction – a little more information is in the discussion lines 358-364 but this needs to come earlier in the introduction too. In the discussion it is not really made clear why previous reports concluded that monoubiquitinate eS10 and uS10 and not polyUb of uS10 is associated with splitting i.e. why the authors could identify polyUb and this was not noticed previously.

2. Line 157-158...what is a final composite map - this is not made clear. Presumably, the two individually processed leading and colliding ribosome maps were merged together on the basis of the most stable disome class and a model was refined into this composite map, but this is not explicitly stated in the main text. It is not clear what percentage of the dimers have the most stable conformation? Especially since it looks like the authors generated the composite maps from all the dimer particles? From Supplementary Table 1, it appears that only one map and model have been deposited, I guess for the composite map and model. It would seem appropriate to deposit maps and models for the individually processed leading and colliding ribosomes?

3. Line 168. I think “conservancy” is generally used in relation to organizations that are involved in protection of the environment. The authors might want to rephrase the sentence. Overall, the conformation observed for the human disome appears to be the same as reported previously for rabbits...not surprising given the high conservation between the two ribosomes. However, the authors describe some additional interface contacts in the human disome that were not observed in the rabbit one, although it is unclear why not, as the elements are conserved. The authors state that the additional contact sites significantly contribute to the overall architecture but since the overall architecture is the same between rabbits and humans, I don't really understand why they believe this? They also state they may have important physiological roles...the manuscript would have been stronger if they could demonstrate this, especially given that the title states that “a distinct human disome collision interface fosters...”...but do they really think that it is these distinct elements that

foster anything...? Otherwise why would the authors then carry out all subsequent experiments with the rabbit system, rather than showing that they get the same or different results in the human one? Perhaps the authors want to tone down their language and change the title, otherwise they should provide some experimental evidence for these statements

4. Figure 3 should be in the supplement, or Figure 2 and 3 should be combined. The direct comparison between human and yeast is not really novel given that the human is the same as rabbit. Also the differences don't appear to be the relevant anyway for splitting since the uS10 is subsequently shown to be the only relevant modification for this and yeast anyway doesn't have the eS10 modification. The emphasis then on the differences, even between the human and yeast seems someone pointless.

5. The data in Figures 4-6 are the highlight of the paper, illustrating that K63-linked polyubiquitin chains are formed by ZNF598 on collided ribosomes are the substrate for RQT activity and not monoubitination of eS10 or uS10. It is only a shame the structures of the K63-linked polyubiquitin collided ribosomes were not presented, rather than the unmodified(?) human collided ribosomes – this would have fitted much better into the paper.

Reviewer #2 (Remarks to the Author):

The authors of the study look into ribosome collisions in RRL and human cells, and explore the ubiquitination of uS10, and eS10 by ZNF598. The study suggests that uS10-ub is important for hRQT ribosome clearance, despite previous data in yeast indicating that eS10 is primarily ubiquitylated and to a lesser extent uS10 on stalled ribosomes.

Overall, the study confirms a lot of discoveries already reported in human cells, RRL in vitro systems, and yeast. The results of the study are interesting for the RQC field, but might not be relevant for the broad readership of the journal. Finally, the experiments performed require additional controls, and often the authors overstated the conclusions of their data.

Specific comments:

1. Fig 1 – can the authors explain why they need to supplement ZNF598 and hRQT to RRL? Is the lysate depleted from RQC components? Is the endogenous ZNF598 inactive? Are the authors overloading the system? Can they show that they observe no ubiquitination on non-stalling reporters even at the supplied dose of exogenous ZNF598?
2. Figure 1F – there is background mono eS10-Ub in a reporter dependent manner (w/o ZNF598) Can the authors explain this result?
3. Fig 1F - for uS10 there is a band just below Ub band that also changes in a reporter dependent manner, and mostly disappears with ZNF598 present. Can the authors explain these results?
4. Fig 4C/D – if ubiquitination is collision dependent, why is there so much ubiquitylation in monosome fractions?
5. Fig 4C/D – can the authors overlay the polysome gradients +/-hRQT. By eye, it looks like a very small fraction of the collided ribosomes get released by hRQT. A better experiment will be to purify collided disomes and release those instead of using full polysomes.
6. Fig 4C/D and E/F– upon careful inspection, I can see faint ubiquitin bands in the 40S fraction for eS10 + hRQT, an observation that goes against the authors' hypothesis. Can the authors explain this result AND provide high exposure gel for these panels?
7. Fig 5C – if the ubiquitin linkage is exclusively K63, why is ubiquitination upon K63only Ub addition less efficiently? Why is there uS10-Ub_{1,2,3} still present in K63R mutant? The author should soften their statements, and clarify that K63 is important, but not the only linkage found.
8. Figure 5E – is there background binding of ASCC3 to Mono Ub and K48?
9. If the authors want to make the conclusion that uS10 ubiquitination is key for hRQT activity, they need the following controls:
 - o Does eS10 lysine mutant still get cleared by hRQT?
 - o Does uS10 lysine mutant block binding and ribosome splitting by hRQT

Typos

1. Typo 4A “westren”
2. Methods ZNF598 from HEK293T cells line 746 “dich”
3. Methods RQC2-FLAG from *S. cerevisiae* line 773 “1 1” typo or unclear
4. Methods Line 912 *in vitro* not italicized
5. Figure S2 legend B “using using”

Point-by-point response to the reviewers' comments

First, we would like to thank both reviewers for their constructive comments and input. We have addressed all your comments in our responses below. The revised manuscript also contains minor modifications throughout to correct typos and increase clarity in some places.

Reviewer #1 (Remarks to the Author):

Ribosome collisions are recognized by the ribosome-associated quality control system and important for rescuing stalled ribosomes in all domains of life. In yeast, Hel2 recognizes collided ribosomes and polyubiquitinates uS10 on the collided ribosome, a mark that is recognized by the RQT complex to promote subunit dissociate. In mammals, ZNF598-mediated ubiquitination of both uS10 and eS10 has been reported, raising the question as to what exactly the signal is that is recognized for ribosome splitting. Here the authors demonstrate that while ZNF598 indeed ubiquitinates both uS10 and eS10, only the polyubiquitination of uS10 (and not monoubiquitination of uS10 or e10) is recognized by the human RQT complex to mediate subunit splitting. Moreover, the authors demonstrate that subunit dissociation requires K63-linked polyubiquitination of uS10 as well as the ATPase activity of ASCC3 of the hRQT complex. These latter points represent the strength of the paper and the biochemical experiments are technically excellent, nicely presented and appropriately interpreted. The authors also present a cryo-EM structure of a human collided ribosome that appears similar to rabbit, but distinct from yeast. It is a little disappointing that it is not the modified collided ribosome or a complex with ZNF598 or RQT. Its also quite confusing in the way that the authors highlight differences with the rabbit system, yet all the biochemistry indicates that these systems are analogous. Nevertheless, overall, I believe its an important piece of work that will be of interest to the scientific community, especially those working on quality control, however, it could benefit from addressing a few mostly minor points.

We appreciate the reviewer's positive comments on our manuscript.

1. Lines 118-119 states that uS10 and eS10 ubiquitination by ZNF598 have been reported previously (reference 4)...however in line 125, it is written as though the detection of polyubiquitination of uS10 is a new finding and makes the mammalian system now analogous with yeast. Presumably the distinction relates to mono- versus poly-, but this is not made clear. In the introduction, the authors write on line 71 that ZNF598 was proposed to preferentially monoubiquitinate eS10 and uS10 in mammals (references 4-6). Given that the whole paper resolves around these modifications maybe some more details need to be provided in the introduction as to why it is just a proposal as opposed to an observation supported by experiments? There are two aspects related to which protein eS10 and uS10 as well as mono- versus polyUb...but these are all jumbled together and its not explained clearly enough for a non-expert in the introduction – a little more information is in in the discussion lines 358-364 but this needs to come earlier in the introduction too. In the discussion its not really made clear why previous reports concluded that monoubiquitinate eS10 and uS10 and not polyUb of uS10 is associated with splitting i.e. why the authors could identify polyUb and this was not noticed previously.

We thank the reviewer for this important point. To make the distinction clear and understandable we changed the sentences starting at line 118 accordingly:

“Consistent with previous reports⁴, we find that both uS10 and eS10 were monoubiquitinated in a ZNF598-dependent manner (Fig 1F). However, while ZNF598 primarily mono- and di-ubiquitinates ribosomal protein eS10¹⁰, we also found that uS10 is in fact polyubiquitinated by ZNF598 analogously to yeast.”

And the sentence in line 70:

“For instance, ZNF598 was reported to functionally monoubiquitinate the eS10 ribosomal protein in mammals^{4,6}, whereas yeast Hel2 functionally polyubiquitinates uS10^{1,9}”

This is an interesting question raised by the reviewer. As a matter of fact, a minor diubiquitination of uS10 was also observed in these studies (e.g. <https://doi.org/10.1016/j.molcel.2016.11.039> Figure 3C). However, the signal was likely too weak as the western blot images are usually cut and no raw data were provided. Therefore, we can only speculate, that this was probably caused by a different experimental setup. Moreover, as discussed in the Discussion (starting at line 437), the ubiquitinated fraction of eS10 is substantially larger than the one of uS10, while apparently not being functional in hRQT-mediated splitting.

2. Line 157-158...*what is a final composite map - this is not made clear. Presumably, the two individually processed leading and colliding ribosome maps were merged together on the basis of the most stable disome class and a model was refined into this composite map, but this is not explicitly stated in the main text. Its not clear what percentage of the dimers have the most stable conformation? Especially since it looks like the authors generated the composite maps from all the dimer particles? From Supplementary Table 1, it appears that only one map and model have been deposited, I guess for the composite map and model. It would seem appropriate to deposit maps and models for the individually processed leading and colliding ribosomes?*

The reviewer's point is correct. We have described the cryo-EM data processing in detail in the Methods section, but a short summary is indeed warranted at this place for better clarity. As requested, we included a clear description of the data processing also in the main text starting at line 158:

"Accordingly, A/P P/E 80S classes (3.9% of all particles) with the most prominent neighboring ribosome density were used to generate a consensus refinement of the disome which was then further refined with focus on the individual 80S ribosomes (while avoiding separate multibody refinements). The individually refined leading and colliding ribosomes were fitted into the consensus refinement to generate a composite map. We reached overall resolutions of 2.9 Å (leading) and 3.2 Å (colliding) for individual ribosomes and 3.0 Å for the composite map (Fig. 2A, Fig. S4)."

This should also clarify, that we did not use multi-body refinement to generate a consensus map and therefore we did not use any arbitrary stable particle subset. Our consensus refinement uses only particles with strong neighbor density and was generated by steps of box extension and re-centering, thereby reaching a resolution of 9 Å. This also explains why there is no further particle subset representing the '*most stable disome class*' as *presumed by the reviewer*. After positioning the individual maps in the well enough resolved consensus map, we could avoid the refinement of individual 80S models in individual maps, as this can easily generate steric clashes in the models when later fitted into the consensus map. This approach is particularly useful when the standard consensus refinement (not substituted by a multi-body refinement) and subsequent focused refinements yield sufficient data quality. Recombining individually refined models may have caused the lack of insight into interface details previously. Maps for the individual monosomes are of course included in the same EMDB entry as additional EM maps defined in the OneDep interface. However, since the molecular models for the two individual 80S and, more importantly, their interface in the disome were refined together, there is only one model which can be deposited.

3. Line 168. *I think "conservancy" is generally used in relation to organizations that are involved in protection of the environment. The authors might want to rephrase the sentence. Overall, the conformation observed for the human disome appears to be the same as reported previously for rabbits...not surprising given the high conservation between the two ribosomes. However, the authors describe some additional interface contacts in the human disome that were not observed in the rabbit one, although it is unclear why not, as the elements are conserved. The authors state that the additional contact sites significantly*

contribute to the overall architecture but since the overall architecture is the same between rabbits and humans, I don't really understand why they believe this?

They also state they may have important physiological roles...the manuscript would have been stronger if they could demonstrate this, especially given that the title states that "a distinct human disome collision interface fosters..."...but do they really think that it is these distinct elements that foster anything...? Otherwise why would the authors then carry out all subsequent experiments with the rabbit system, rather than showing that they get the same or different results in the human one? Perhaps the authors want to tone down their language and change the title, otherwise they should provide some experimental evidence for these statements

We thank the reviewer for this valid point. The sentence in line 174 now states:
"This conserved architecture is further illustrated by only very small positional differences we observed when comparing the structures of rabbit and human disomes (Fig 2D)."

We believe that the additional contact sites significantly contribute to the overall architecture, since they are so prominent and extensive. We especially highlight how the previously undescribed contact of ES6c is one of the main determinants of the inter-ribosomal orientation in mammals. To describe this in other words: we now define for the first time the structural determinants of these highly similar disome assemblies. As far as we can tell from re-analyzing the rabbit disomes, the same contacts are present in both species but were simply not described/discussed in the text or figures of the previous publications. We therefore agree with the reviewer that the overall architecture and functionality of both rabbit and human disomes are highly conserved and equivalent, therefore warranting their use in biochemical experiments that mix rabbit and human components.

We also agree with the reviewer that we should reduce the speculation in the Results and in the Title. We have now removed the speculative conclusion of the first structural chapter, which now states:

"Nonetheless, the additional contact sites we describe here significantly contribute to the overall architecture of mammalian collided disomes."

According to the reviewer's remarks we have now also modified the title exactly as suggested:

*"A distinct **mammalian** disome collision interface **harbors** K63-linked polyubiquitination of uS10 to trigger hRQT-mediated subunit dissociation"*

4. Figure 3 should be in the supplement, or Figure 2 and 3 should be combined. The direct comparison between human and yeast is not really novel given that the human is the same as rabbit. Also the differences don't appear to be the relevant anyway for splitting since the uS10 is subsequently shown to be the only relevant modification for this and yeast anyway doesn't have the eS10 modification. The emphasis then on the differences, even between the human and yeast seems someone pointless.

To our best knowledge, neither the experimental evidence for the highly conserved architecture of the rabbit and the human disome nor the direct comparison of either mammalian disome with the yeast one was published before. Therefore, both these aspects of our study represent novel findings. It is correct, that both rabbit and human disomes could have been used equivalently for this comparison. In any case, the mammalian and yeast disome assemblies are distinct as illustrated by the respective 18° rotation and its effect on the interface. This can be important for splitting, since we observe a substantial difference in RQT splitting efficiency between human and yeast and our proposed RQT splitting mechanism suggests a role of a stable head-to-head contact (<https://doi.org/10.1101/2022.04.19.488791>). Moreover, this structural difference could potentially cause the distinct collision-dependent ubiquitination patterns and thereby might affect all the downstream steps. In this regard, the eS10 ubiquitination which is absent in yeast (as correctly mentioned by the reviewer) can have a specific role in mammals. Therefore, we highlight in the title of the manuscript that mammalian disomes generate a

distinct interface and followed the reviewer's suggestion to combine parts of the two figures into one instead of moving these important results entirely into the Supplement.

5. *The data in Figures 4-6 are the highlight of the paper, illustrating that K63-linked polyubiquitin chains are formed by ZNF598 on collided ribosomes are the substrate for RQT activity and not monoubitination of eS10 or uS10. It is only a shame the structures of the K63-linked polyubiquitin collided ribosomes were not presented, rather than the unmodified(?) human collided ribosomes – this would have fitted much better into the paper.*

We agree with the reviewer that seeing the structure of polyubiquitin-modified disomes would improve our understanding of the whole process. Unfortunately, given the conformational flexibility of polyubiquitin chains *per se* (<https://doi.org/10.1016/j.jmb.2009.07.090>) as well as the flexible tails of ribosomal proteins that are polyubiquitinated, this represents a major challenge. Indeed, even when extensively polyubiquitinated *in vitro*, ubiquitin modifications remained invisible in the recent structural study of RQT splitting in yeast (<https://doi.org/10.1101/2022.04.19.488791>). Nonetheless, we found in yeast that the overall disome conformation was not altered by ubiquitination and it is highly unlikely to have an effect on disome conformation in the mammalian system.

Reviewer #2 (Remarks to the Author):

The authors of the study look into ribosome collisions in RRL and human cells, and explore the ubiquitination of uS10, and eS10 by ZNF598. The study suggests that uS10-ub is important for hRQT ribosome clearance, despite previous data in yeast indicating that eS10 is primarily ubiquitylated and to a lesser extent uS10 on stalled ribosomes. Overall, the study confirms a lot of discoveries already reported in human cells, RRL in vitro systems, and yeast. The results of the study are interesting for the RQC field, but might not be relevant for the broad readership of the journal. Finally, the experiments performed require additional controls, and often the authors overstated the conclusions of their data.

We appreciate the valuable comments on our manuscripts. We would like to emphasize that we reported that the polyubiquitylation of uS10 is required for the RQT-mediated splitting in yeast, and that eS10 ubiquitination was NOT detected in yeast. What the reviewer describes are results previously reported for the mammalian system. In this study, we show that K63-linked polyubiquitination of uS10 is crucial for hRQT-mediated subunit dissociation in mammalian cells, indicating the conservation of the RQC-triggering step. We believe that this study is important to understand the mechanism of the quality control that recognizes and eliminates the aberrant translation to maintain the protein homeostasis in mammals.

Specific comments:

1. *Fig 1 – can the authors explain why they need to supplement ZNF598 and hRQT to RRL? Is the lysate depleted from RQC components? Is the endogenous ZNF598 inactive? Are the authors overloading the system? Can they show that they observe no ubiquitination on non-stalling reporters even at the supplied dose of exogenous ZNF598?*

We thank the reviewer for these questions. We did not deplete the RQC components nor supplemented hRQT to RRL in Figure 1. The endogenous ZNF598 was not sufficient for clear visualization of ubiquitinated uS10 or eS10. Only residual ubiquitination could be observed without additional ZNF598 likely due to the presence of the endogenous downstream effectors and deubiquitinases. Nonetheless, this indicates that the endogenous ZNF598 is indeed active. According to the addition of ZNF598, the ubiquitination of uS10 and eS10 increased.

To convince the reviewer that also the exogenous ZNF598 is indeed acting on collided ribosomes and not on reporters that do not induce collisions, we performed additional experiments to show the difference in ubiquitination on *Rz* or truncated mRNA constructs (no collisions) and XBP1u or poly(A) (collisions) constructs at different doses of exogenous

ZNF598 (250 nM; Figure 1a, lanes 4-5 and Figure 1b, lanes 5-8, 12-14).

The rebuttal figure 1

2. Figure 1F – there is background mono eS10-Ub in a reporter-dependent manner (w/o ZNF598) Can the authors explain this result?

As discussed throughout the manuscript, the ubiquitination efficiency of eS10 is much higher than the one of uS10 while not being functionally important for RQT splitting. This is reflected by the faint mono-ubiquitinated eS10 background, indicating that endogenous ZNF598 ubiquitinates eS10 of ribosomes stalled by the *XBP1u* and poly(A) stallers. Importantly, eS10 monoubiquitination seems to be present as a general background as demonstrated in our experiments in human HEK293T cells even without any added stall-inducing anisomycin (Fig S1C). We now discuss this in the manuscript, starting at line 118 & 413.

3. Fig 1F - for uS10 there is a band just below Ub band that also changes in a reporter dependent manner, and mostly disappears with ZNF598 present. Can the authors explain these results?

We agree that this band does not seem to size-wise correspond to monoubiquitination. We can only speculate that this band represents another modification of the tagged uS10. Regardless, there is no clear relation of this band in Figure 1F with the ubiquitination or splitting of collided ribosomes visualized throughout the manuscript.

4. Fig 4C/D – if ubiquitination is collision dependent, why is there so much ubiquitylation in monosome fractions?

We thank the reviewer for this question. There is a significant body of published experimental evidence showing that ZNF598/HeI2 mediated ribosome ubiquitination is indeed collision dependent (see e.g. PMIDs: 30293783, 30609991, 28757607, 33338396, 32615089 and 32579943). There are two plausible explanations of the polyubiquitinated uS10 in monosomes fractions. One is that ubiquitinated monosomes cannot be further dissociated by the RQT complex (DOI: 10.1101/2022.04.19.488791). Another one is the reassembly of the 60S and the ubiquitinated 40S product of RQT complex-mediated subunit dissociation. The third possibility is that the ubiquitinated 80S ribosomes in the polysomes are dissociated during the purification and incubation for the splitting reaction. To clarify this, we now discuss these possibilities in the discussion starting at line 423.

5. Fig 4C/D – can the authors overlay the polysome gradients +/-hRQT. By eye, it looks like a very small fraction of the collided ribosomes get released by hRQT. A better experiment will be to purify collided disomes and release those instead of using full polysomes.

We thank the reviewer for this valid point. As higher-order collided ribosomes were shown to be a better substrate for RQT and RQC (Matsuo et al., 2020), we purified the polysome fractions (heavier than trisomes) after the ZNF598-mediated ubiquitination and subjected them to the hRQT-mediated splitting (Figure S6). The polysomes with the polyubiquitinated uS10 are efficiently dissociated into the ubiquitinated 40S and 60S subunits. In addition, polysomes with the polyubiquitinated eS10 are dissociated into the ubiquitinated 40S but only with low efficiency. These strongly suggest that the collides ribosomes with the polyubiquitinated uS10 are efficiently dissociated into the subunits by the hRQT complex. We integrated these results in the new Figure S6 and in the results starting at line 252.

6. Fig 4C/D and E/F– upon careful inspection, I can see faint ubiquitin bands in the 40S fraction for eS10 + hRQT, an observation that goes against the authors' hypothesis. Can the authors explain this result AND provide high exposure gel for these panels?

As discussed throughout the manuscript, the ubiquitination efficiency of eS10 is much higher than the one of uS10 while not being functionally important for RQT splitting. The difference between the minimal band of monoubiquitinated eS10 and the strong signal for polyubiquitinated uS10 (which is ubiquitinated much less efficiently as seen in these panels) is striking. Importantly, as already mentioned above, the eS10 monoubiquitination seems to be present as a general background as demonstrated in our experiments in human HEK293T cells without any added stall-inducing anisomycin (Fig S1C).

As requested by the reviewer, we provided high-exposure western blots for these panels, showing the mono-ubiquitinated bands in the 40S fraction for eS10+hRQT more clearly. The plausible explanation for this is that the hRQT complex efficiently dissociates collided ribosomes with polyubiquitinated uS10, which to some extent coincides with the monoubiquitinated eS10. We now discuss this in the revised discussion starting from line 418.

7. Fig 5C – if the ubiquitin linkage is exclusively K63, why is ubiquitination upon K63only Ub addition less efficiently?

We thank the reviewer for this valid question. This most likely reflects the lower efficiency of the ubiquitination reaction using the K63-only mutant in comparison with WT ubiquitin. There are many examples showing that polyubiquitination reaction with K63only ubiquitin is less efficient than the one of the wild-type (PMIDs 33339882 and 25847972).

Why is there uS10-Ub1,2,3 still present in K63R mutant? The author should soften their statements, and clarify that K63 is important, but not the only linkage found.

There are two ubiquitination sites in uS10, K4 and K8. Therefore, the two major bands of the ubiquitinated uS10 (uS10-Ub and uS10-Ubx2) represent single (at K4 or at K8) or double monoubiquitinated uS10 (at both K4 and K8). In Figure 5A-B, only the uS10-Ub and uS10-Ubx2 were detected, suggesting that only (di-)monoubiquitination of uS10 is possible with the Ub-K63R mutant. In Figure 4C, the faint Ubx3 band would suggest that another linkage-type ubiquitin could be added on uS10 with low efficiency. However, the same background band is also present in the polysome input and in the K0 control, which does not allow ubiquitin linkage. Therefore, this band is unlikely to represent the suggested Ubx3. Moreover, the differences in ubiquitination efficiency between the K63R and the K63only or the wt ubiquitin are striking.

8. Figure 5E – is there background binding of ASCC3 to Mono Ub and K48?

The faint background band the reviewer seems to be referring to does not correspond to mono-Ub or K48-linked tetra-Ub. In lanes 1-2, this faint background band is clearly of a different size to either Mono-Ub (Compare lanes 1 and 4) or Tetra-Ub(K48) (Compare lanes 2 and 5). Figure 4D (the previous Figure 5E) indicates that ASCC2 subunit

of the hRQT complex preferentially binds to the K63-linked polyUb chain. These results are consistent with previous studies demonstrating that ASCC2 binds to K63-linked but not K48-linked ubiquitin chains via its CUE domain (Brickner et al Nature 2017; Lombardi et al JBC 2022). To demonstrate the linkage-specific interaction of the hRQT complex containing ASCC2, we modified the binding buffer and found no background signal of the hRQT complex binding to Mono-Ub and Tetra-K48 (The revised Fig. 4E). In addition, we also demonstrated that hRQT complex binds to the K63-linked polyubiquitin chain (The revised Fig. S7D) and found no background signal in the hRQT complex binding to Ub (The revised Fig. 4D, lane 3 and 4E, lane 4).

9. If the authors want to make the conclusion that uS10 ubiquitination is key for hRQT activity, they need the following controls:

o Does eS10 lysine mutant still get cleared by hRQT?

o Does uS10 lysine mutant block binding and ribosome splitting by hRQT?

We appreciate this suggestion as we have previously used analogous controls in the easier to handle yeast system. Similar experiments that can be done in human cells in a relatively short amount of time were already done using simply overexpression of mutant uS10 and eS10 proteins in human cells (doi: 10.1038/ncomms16056, 10.1016/j.molcel.2016.12.026, <https://doi.org/10.1016/j.molcel.2016.11.039>). However, while Juszkiwicz and Hegde observed only a partial effect in uS10 K4R/K8R double mutant, Sundaramoorthy, et al. observed that both uS10 and eS10 mutants displayed enhanced readthrough of a poly(A) sequence. Finally, Garzia et al. conclude that only eS10 K138R, and the double mutant K138R/K139R partially impaired RQC for the (AAA)₂₀ staller, while uS10 mutants had no effect. Therefore, transient overexpression of mutant ribosomal proteins does not seem to provide clear answers.

As a consequence, providing these controls in a conclusive manner would require extensive time and effort, which we estimate to be at least 6-12 months. We would have to perform the following experiments:

1. Generate the corresponding bi-allelic human cell lines of HEK293/HeLa cells expressing only uS10-KR or eS10-KR mutants using the CRISPR/Cas-System followed by extensive clone screening.
2. Establish and optimize cell-free translation extracts from these human cell lines for the *in vitro* experiments.
3. Optimize the *in vitro* ZNF598-mediated ubiquitination and RQT complex-mediated subunit dissociation reaction in this human system. This could then be followed by analysis through sucrose-gradient ultracentrifugation and Western blotting.

Steps 2 and 3 would be critical, as both this and previous studies use the rabbit reticulocyte lysate instead of human cell extract for a good reason. It is crucially important to be able to work with sufficient amounts of material to optimize the *in vitro* translation, ubiquitination and splitting reactions. Using the generated human cell lines to optimize and perform the desired experiments would require both extensive labor and significant resource investments over an extended time frame.

Taken together, while these controls would be useful to further reinforce the conclusion that uS10 polyubiquitination is key for hRQT activity and eS10 polyubiquitination also contributes to the dissociation but the polyubiquitination efficiency is significantly lower than that of uS10, we would argue, that this is already sufficiently proven for two main reasons in our current manuscript:

- There is a substantial body of experimental evidence from at least three different research groups, showing that eS10 and uS10 are the only possible functional

targets of ZNF598 ubiquitination to initiate RQC in mammals. It is therefore highly unlikely that any unknown ubiquitination target besides uS10 or eS10 would be driving the downstream steps.

- We can clearly show that polyubiquitination is necessary and that the polyubiquitinated uS10 and eS10 as a product of the hRQT-mediated splitting as documented in the revised Figure S6.

Typos

1. *Typo 4A “westren”*
2. *Methods ZNF598 from HEK293T cells line 746 “dich”*
3. *Methods RQC2-FLAG from S. cerevisiae line 773 “1 1” typo or unclear*
4. *Methods Line 912 in vitro not italicized*
5. *Figure S2 legend B “using using”*

We have corrected these typos.

REVIEWERS' COMMENTS

Reviewer #2 (Remarks to the Author):

All my concerns have been addressed in the revision.